



# Central-Pacific surface meteorology from the 2016 El Niño Rapid Response (ENRR) field campaign

Leslie M. Hartten[1,2], Christopher J. Cox[1,2], Paul E. Johnston[1,2], Daniel E. Wolfe[1,2], Scott Abbott[2], H. Alex McColl[1,2,3]

5   [1]Cooperative Institute for Research in Environmental Science (CIRES), University of Colorado, Boulder, 80309-0216, USA
[2]NOAA/Earth System Research Laboratory, Physical Sciences Division, Boulder, Colorado, 80305, USA
[3]currently at Berthoud, Colorado, 80513, USA

*Correspondence to*: Leslie M. Hartten (Leslie.M.Hartten@noaa.gov)



**Abstract.** During the early months of the 2015/16 El Niño event, scientists led by the Earth System Research Laboratory's Physical Sciences Division conducted NOAA's El Niño Rapid Response (ENRR) Field Campaign. One component of ENRR involved in-situ observations collected over the near-equatorial East-Central Pacific Ocean. From 25 January to 28 March 2016, standard surface meteorology observations, including rainfall, were collected at Kiritimati Island (2.0° N, 157.4° E) in

support of twice-daily radiosonde launches. From 16 February to 16 March 2016, continuous measurements of surface meteorology, sea surface temperature, and downwelling shortwave radiation were made by the NOAA Ship *Ronald H. Brown*. These were largely done in support of the four to eight radiosondes launched each day as the ship travelled from Hawaii to TAO buoy locations along longitudes 140° W and 125° W and then back to port in San Diego, California. The rapid nature of these remote field deployments led to some specific challenges in addition to those common to many surface data collection

efforts. This paper documents the two deployments as well as the steps taken to evaluate and process the data. The results are two multi-week surface meteorology data products and one accompanying set of surface fluxes, all collected in the core of the east-central Pacific's extremely warm waters. These data sets, plus metadata, are archived at the NOAA's National Centers for Environmental Information (NCEI) and free for public access: surface meteorology from Kiritimati Island (doi:10.7289/V51Z42H4); surface meteorology and some surface fluxes from the NOAA Ship *Ronald H. Brown*

(doi:10.7289/V5SF2T80; doi pending, available at http://accession.nodc.noaa.gov/0167875).

**Keywords.** El Niño, surface meteorology, surface fluxes, tropical Pacific, central Pacific, Kiritmati Island, ship observations

## 1 Introduction

In June 2015, the weak El Niño conditions that had existed since March 2015 were strengthening and forecasters were

confident that they would continue to do so through winter 2015/16 (Climate Prediction Center (CPC) and International Research Institute for Climate and Society, 2015). The United States' National Oceanic and Atmospheric Administration (NOAA) wanted to quickly address the unfolding event, which had the potential for large domestic impacts, particularly on the country's Pacific coast, and thus developed the El Niño Rapid Response (ENRR). The Earth System Research Laboratory's (ESRL) Physical Sciences Division (PSD) led the design and implementation of one component, the

ENRR Field Campaign (Dole et al., 2017). Among the many assets put into play were surface meteorological instruments on Kiritimati (pronounced "Christmas") Island and aboard the NOAA Ship *Ronald H. Brown*. The primary purpose of the surface meteorological measurements was to provide initialization data for the radiosondes launched twice a day from Kiritimati and four to eight times per day from the NOAA Ship *Ronald H. Brown* (Hartten et al., 2017a). However, the high temporal resolution of the surface measurements in these remote locations makes them useful in their own right. The quick deployment

and remote locations led to several data challenges which needed to be overcome both for the sake of creating research-quality radiosonde data sets and to enable the independent value of the surface data to be more fully realized. This article documents the data collection, the problems identified and corrections applied after the field phase, and the resulting data sets.



## 2 Instrument Specifications and Siting

### 2.1 Kiritimati Island, Republic of Kiribati[1]

Kiritimati is one of the Line Islands, a chain of islands and atolls lying across the equator south of Hawaii. Its land area of 321 km$^2$ makes it the world's largest coral atoll (Scott, 1993); its lagoons cover a similar area (Fig. 1a). Much of it lies a few

meters above sea level, with its highest point only 13 m ASL. Kiritimati's 6,456 inhabitants as well as all visitors live in four towns on the northern side of the island (Morate, 2016). The Kiribati Meteorological Service maintains and staffs an office at Cassidy International Airport along the northeast-facing coastline; observers record surface conditions hourly and report them under WMO station ID 91490. Its position in the equatorial central Pacific, coupled with its relatively large land area and advanced infrastructure, have made Kiritimati Island the site of choice for many previous scientific efforts, including the Line

Islands Experiment (Zipser, 1970); the multi-year deployment of wind-profiling radars (Gage et al., 1991); climatological studies using coral (Evans et al., 1999); and the Pacific Atmospheric Sulfur Experiment (PASE; Conley et al., 2009).

ENRR operations were based at the Captain Cook Hotel, which is located on the northeast-facing coast about 6 km from the airport (Fig. 1a). This location was chosen because it was familiar to some of our staff; had experience hosting scientific field

work; had 60 Hz, 120 V power and internet access; and had enough driveable paths, staff presence, and public lighting to provide a level of safety during evening radiosonde launches. Field staff lived in a two-unit bungalow (Fig. 1b). One unit doubled as office space, housing the computers and radiosonde equipment, and for a few weeks supplied power to the outdoor instruments. The bungalow was usually air-conditioned with the temperature set to 24-26 °C. The site was located at (2.01° N, 157.40° W), and upper-air observations were transmitted to the Global Telecommunication System (GTS) using

"CXENRR" as a station name.

On 25 January 2016 UTC, the first day of operations, the surface meteorological instruments were attached to a tripod (Fig. 2) located about 14 m northeast of the concrete pad seen in Fig. 1b. Table 1 contains information about the surface meteorology instruments deployed, the variables measured, and recording details. A solar panel was also attached to the tripod; it was used

to charge the battery that ran the Campbell datalogger. This initial location was further from our bungalow than expected, and we had no cable long enough to connect directly to the computers inside the bungalow, so surface conditions could not be monitored in real-time and an observer had to go out and read instantaneous surface values directly off the Campbell datalogger during the radiosonde ground check and initialization. Within a few days staff realized that the initial location was somewhat obstructed by the two nearby bungalows (Fig. 1b, Fig. 3a), which were approximately 12 m apart from each other. On 5

February the tripod was moved northeast about 10 m, to a location near the top of the beach (Fig. 1b, Fig. 3b) that was further from the bungalows, and a newly arrived 30 m cable was run between the station and the computers in our bungalow. This

---

[1] The country name is pronounced "KEER-eh-bahss".





second location was also somewhat obstructed (Fig. 3b), with a tree about 12 m to the west-northwest and the two bungalows about 15 m to the southwest and 6 m to the south. However, it was believed superior because of its more open setting and long fetch over a 180° sector. This setup was maintained through 28 March 2016, the last day of CXENRR operations.

### 2.2 NOAA Ship *Ronald H. Brown*

The NOAA Ship *Ronald H. Brown* is a Global Class research vessel in service since 1997 as part of the NOAA Marine Operations Center – Atlantic (MOC-A) fleet. The vessel is 83.5 m long with a 16.2 m beam and has a cruising speed of 11 knots. When the ENRR field campaign was being planned, the ship was already scheduled to cruise from Hawaii to San Diego during February and March 2016 to conduct routine maintenance of buoys that are part of the Tropical Atmosphere Ocean (TAO) array along 8° N to 8° S meridional transects at 140° W and 125° W (Fig. 4). Because the scheduled timing and path

of the ship was well-suited for the objectives of the ENRR campaign, three scientists from ENRR were deployed with the ship to acquire soundings during the full duration of the cruise. These were transmitted to the GTS as TEMP MOBILE messages using the station name "WTEC".

The ENRR surface meteorology and oceanographic observations were collected by the some of the ship's standard

instrumentation suite (Fig. 5). Technical information about the instruments deployed, the variables measured, measurement and recording details, and operation notes are given in Table 2. As at Kiritimati, our primary interest in the surface meteorological data collected during the cruise was for the radiosonde ground check and initialization. The radiosondes were launched four to eight times per day from the main deck of the ship, and the radiosonde ground-check equipment was located indoors on the main deck of the ship approximately 3.8 m above the ship's designated water line.

**3 Observational Issues and Post-Deployment Data Correction**

Both sites had difficulties with some of their surface measurements. Some problems were almost immediately apparent; others became known as the campaign unfolded or upon post-deployment review. The problems and the methods used to correct them were different for each site and are described below. Only those measured quantities that required detailed post-deployment analysis are discussed in subsections; all other information is given in the main site sections.

**3.1 Kiritimati Island**

Issues in the original Kiritimati Island surface data fall into five categories: instrument calibration issues; instrument failures; non-representativeness of the data caused by poor siting; changes in the data acquisition methods and archival timing; and gaps in the data. Some of these issues were consequences of the "Rapid Response" aspect of the project. Because there is only one international flight to Kiritimati Island each week and the campaign was pulled together in only a few months, there was

no time for a site survey prior to the start of fieldwork. This limited the choices of available sites and led to some of the



shipped equipment being our "best guess" at what would be needed. It also meant that we often had to make do with what was available on the island, or wait a week or more until something could be shipped.

The "Rapid Response" also affected the exact instruments deployed, which had to be those available on short notice from
PSD's instrument pool. Questions about the calibration of some of them arose soon after the campaign started, which led us to develop a practice of placing the initialized radiosonde atop the instrument box for an extended period before launch. Radiosonde initialization, which was done inside the air-conditioned bungalow, includes a ground check procedure that uses high-precision and high-accuracy temperature and humidity sensors to calibrate each radiosonde. Setting the radiosonde in the shade of the surface station's solar panel (Fig. 2b) allowed the radiosonde to equilibrate to the outside environment; it also
provided minutes of co-located pressure, temperature, and humidity data from the radiosonde and the surface instruments. Details of how we used these coincident measurements to refine the surface observations are discussed in the relevant subsections below.

There were two instrument failures during the experiment. During the second week of operations it became clear that the
temperature/humidity probe had failed. Fortunately, we had brought a second unit with us. We swapped the sensors on 6 February and finalized the replacement sensor's settings on 7 February 2016 at 03:41 UTC; it operated well for the rest of the campaign. On 15 March 2016, at 11:12 UTC, the solar panel providing power for the data logger failed. It took about 18 hours to diagnose the problem, find a replacement power supply and install it, and restart the surface station.

The surface observations were critical for the initialization of the radiosondes, so the surface station had to be located near the radiosonde release location, i.e. close to the bungalow housing the radiosonde data system. The surface station was originally set up on a grassy area northeast of the concrete pad. This location looked good at first, but after a few days of operations the staff realized that the anemometer sometimes spun in circles, indicating swirling winds; that the range of recorded (2-minute) wind directions was quite narrow; and that recorded winds were sometimes inconsistent with the values reported by the
radiosondes after they had risen above the bungalows and trees. The surface station was moved to a better location, at the edge of the beach and further from all the bungalows, on 5 February 2016 (21-22 UTC).

On 11 February 2016, at 04:05:43 UTC, four changes were made to the data logger program. The execution interval, which determines how often each instrument is read, was changed from 2 seconds to 1 second. The internal offset for the barometer
was changed from 599.5 to 399.5, so that pressure would be reported with a resolution of 0.01 hPa instead of 0.1 hPa. The averaging time was changed from 2 minutes to the more standard 1 minute. All these changes were made to give better data to help with radiosonde surface initialization and with data evaluations.



The failure of the T/RH sensor, the station move, the change to the data logger software, and the failure of the data logger power supply all caused gaps in the data (Table 3). We have not attempted to interpolate across any of the data gaps, nor have we tried to interpolate the two-minute data to one-minute values. Instead we have carefully examined data recorded before and after instrument moves, failures, and replacements, and have replaced suspicious data with flags.

There is one final general note about the surface meteorology data. The radiosonde system maintains time very accurately, but it does so by getting the time from GPS satellites. The Vaisala Digicora software sets the computer clock to GPS time, which was 17 s ahead of UTC time during ENRR, and the Campbell software sets the data logger clock to the computer's time. For the final Kiritimati surface data set, all times have been shifted by 17 s, so that all observations are reported in UTC.

**3.1.1 Surface Pressure**

The radiosonde ground check procedure requires an external pressure reading to provide a calibration check and correction to the radiosonde pressure sensor. This was provided by the Vaisala PTB101B pressure sensor mounted at 3.6 m ASL on the instrument tripod. The barometer was installed with a Vaisala SPH10 static pressure head to minimize wind induced errors. During the first month of the experiment, the staff noted pressure differences between the PTB101B and both the radiosondes

(Fig. 6) and other barometers. The PTB101B was checked against a high-quality calibrated standard after returning to Colorado. The PTB101B read 0.64 hPa too high, so in the final data set all the surface pressure data was decreased by 0.64 hPa to account for this calibration offset.

**3.1.2 Relative Humidity and Temperature**

The radiosonde ground check procedure serves to calibrate each radiosonde's humidity sensor, so it became obvious very early

in the project that there were problems with the station's original HMP45C humidity probe. These were not entirely solved by the replacement HMP45C. Figure 7a shows the HMP45C humidity readings with the co-located radiosonde data before the 14 February 2016 12 UTC launch. This was a very typical situation, with the HMP45C consistently reporting lower humidity than the radiosonde. Analysis of all co-located data collected after the HMP45C was replaced shows that the mean difference between the HMP45C and the co-located radiosonde humidity was 5.1% (Fig. 7b). We have therefore corrected

the surface relative humidity for this bias by adding 5.1% to all reported relative humidity values that were considered good. The final values are shown in Fig. 8b.

During post-deployment evaluation of our data, we also compared the one-minute HMP45C temperature with one-minute averages of radiosonde temperatures collected while the radiosonde was co-located with the instrument tripod. All

30 comparisons are with the replacement sensor. The results (Fig. 9) show that the mean temperature difference between the radiosonde temperatures and those from the HMP45C is less than ±0.1 °C. This is better than expected, given the differences



of shielding and aspiration. We concluded that the temperature probe data, except for the initial deployment period when there was a bad probe, could be used as recorded in the field. Figure 8a shows the final time series.

### 3.1.3 Winds

Several hours after initial setup the base of the anemometer was aligned to face south using a Brunton hand-held compass.

The user was standing within a few meters of the tripod and the anemometer was 3.88 m above ground level. Wind observations prior to this are of speed only. The alignment was repeated when the station was relocated. On 28 March 2016 the on-site observers used a hand-held compass and string to determine that the anemometer was aligned at 168° relative to magnetic north, within a ±3° range. Since the declination for our site is 9.03° east of north (National Centers for Environmental Information, 2016), the values recorded on site will show a 180° wind direction when the

wind is 168°+9°=177°. We have therefore subtracted 3° from the recorded wind directions to correct for this bias. This correction was applied to the entire record, since the alignment method was the same during initial deployment and after the move. The measured wind speeds are the best estimate we have of the actual wind speeds, and we have not changed them. The final time series is shown in Fig. 10.

As mentioned above, neither the initial nor the final location of the surface instruments was ideal from a wind perspective. The bungalows were approximately 6 m high, and no place near our launch site qualified as "open terrain", i.e. with "the distance between the anemometer and any obstruction … at least 10 times … the height of the obstruction" (World Meteorological Organization, 2013, Part III Section 3.3.5). Therefore, even after the move there were still blockage issues affecting the measured winds. However, moving the station clearly increased the range of wind directions and speeds we

observed. We have analysed the corrected winds over several time periods, calculating various statistics from the high-resolution time series and also evaluating the directions in both 5° and 10° bins[2]. Before the move, the mean wind speed was 2.23 m s$^{-1}$ with a mean direction of 22.5°; 71% of the winds came from 0°–50° (Fig. 11, left), with very small secondary clusters from the south (160°–180°, 5%) and southwest (230°–275°, 7%). Only 3.4% of the measured winds were from the 50°–160° sector, which is surprising considering Kiritimati's location in the trade wind region. After the move, the mean wind

speed was 4.52 m s$^{-1}$ and the mean direction 61.9°; 85% of the winds came from 15°–100° (Fig. 11, right), with large clusters at 50°–70° (28%) and 80°–95° (19%). Winds from the remainder of the 50°–160° sector, i.e. from 100°–160°, were uncommon (4%), while extremely small but distinct secondary clusters were still present from the south (175°–195°, 2%) and southwest (240°–265°, 1%).

---

[2] The 10° bins provide a representative visual summary of the slightly more nuanced view the 5° bins show, so we present figures using 10° bins accompanied by quantitative discussion from the 5° binned data.





Statistical analysis of the station move's impact, e.g. whether or not the winds are different before and after the move, is complicated both by the very unequal lengths of the "before" and "after" periods (and the attendant possible large-scale meteorological variations) and by the change in averaging period on 11 February 2016. We have therefore formally compared the 5.25 days' worth of 2-minute winds available after the move with 5.25 days' worth of winds immediately before the move.

Our analysis employs circular statistics; it follows Fisher (1995) and relies heavily on MATLAB routines described by Berens (2009). Rose diagrams from these equal before and after periods are shown in Fig. 12, key statistical quantities in Table 4, and calculation details in Appendix A. The before and after wind roses show many of the same features noted in Fig. 11, and Fig. 10 shows that each 5.25 day period incorporates variability similar to the longer "before" and "after" time series. The null hypothesis that the mean directions are the same is to be rejected if the test statistic $Y_r$ is too large when compared to the

upper $100(1-\alpha)\%$ point on a $\chi^2_\nu$ distribution with $\nu = r - 1$ degrees of freedom. In this case $r = 2$, and $Y_2$ far exceeds the threshold needed to reject the null hypothesis at the 0.1% significance level.

### 3.2 NOAA Ship *Ronald H. Brown*

Post-processing and quality control of the surface data recorded on the NOAA Ship *Ronald H. Brown* made use of the ship's occasional proximity to TAO buoys that were actively collecting data (generally the buoys were offline for maintenance when

the ship was nearby), and also took advantage of equipment associated with our radiosondes launches. Before launch the Vaisala RS92-SGP radiosondes were put through a ground-check process inside an air-conditioned space. Because the final surface meteorology data would be needed for the post-processed radiosonde data, some of these ship observations have been adjusted to the height of the radiosonde ground-check unit (3.8 m ASL).

### 3.2.1 Station Pressure

Atmospheric pressure was measured with a Vaisala PTB 330 which had been calibrated on 29 December 2015. While the barometer was located on the ship's bridge (15.24 m ASL) the pressure reported in the data set has been adjusted to 3.8 m, the height of the radiosonde ground-check unit, via the hypsometric equation.

The ship's pressure sensor is mounted on the port side of the bridge. Airflow distortion caused by the superstructure of the

ship can result in localized pressure anomalies. We compared the ship pressure with the pressure observed by the radiosondes while the radiosondes equilibrated outside on the fantail of the ship (see Hartten et al. 2017a, in preparation). This analysis revealed a bias in the ship pressure as function of relative wind direction that is approximately 0.007 hPa per degree of relative wind direction (r = -0.46, $p < 0.001$): positive when the winds are incident upon the port side, negative when they are incident upon the starboard side, and near zero when the relative wind direction is aligned along the ship's length. However, while a

correction improves the overall statistical bias of the data set, it introduces discontinuities into the time series when applied to these high temporal resolution data. Therefore, the uncertainty is reported here and a correction was applied to the pressures



that were input as surface observations for the soundings, but no such correction was applied to the 1-minute average pressure in the surface data set.

The TAO buoy at 0° N, 140° W (dmb166) reports atmospheric pressure at the top of every hour using a Paroscientific MET1
pressure sensor mounted at 3 m (National Data Buoy Center, 2010a,b).  Unfortunately, when the ship arrived on location we discovered that dmb166 was damaged, so no data are available during our approach.  The closest overlapping pressure observations are from after the buoy was repaired, on 28 and 29 February 2016 while the ship was traveling south but still within several hundred kilometers of the buoy.  Comparing the ship pressure, adjusted to the buoy pressure sensor height of 3 m ASL, and the buoy pressure during this time (Fig. 13) shows that the two were in close agreement from when the buoy
pressure becomes available until around 06 UTC on the 29[th] when the two measurements began to diverge systematically, likely because of the large distance between the sensors (more than 350 km after 07 UTC on the 29[th]). From 14 UTC on the 28[th] through 07 UTC on the 29[th] the mean and standard deviation of the difference (ship minus buoy) is -0.08 hPa ± 0.24 hPa.

### 3.2.2 Temperature

Atmospheric thermodynamic temperature was measured at 1 Hz by an IMET Rotronic MP-101A mounted on the bow mast at
approximately 15 m ASL.  The sensor, which has been modified by Woods Hole Oceanographic Institution (WHOI; 2010a, b), was last calibrated on 26 July 2014.  Most TAO buoys record temperature, so the ship's air temperature can be validated against many measurements made away from the superstructure of the ship. The buoys record temperature at 3 m ASL using Rotronic MP-101A temperature-humidity probes (National Data Buoy Center, 2010a, b). Buoy measurements are recorded every 10 minutes based on 2-min averages of 2 Hz measurements around the sample time. Figure 14 shows a comparison
between the temperature from the bow mast, averaged to match to buoy sampling, and the temperatures reported from select buoys. Comparisons are only made when the ship was within 0.1º latitude and 0.1° longitude (~10-15 km) of the buoy, i.e. on 29 occasions involving four buoys (at (9º N, 140º W), (5º N, 140º W), (5º S, 140º W), and (8º N, 125º W)). The difference between the ship and buoy temperatures (ship minus buoy) is -0.09 °C ± 0.25 °C, which is within the uncertainty in the sensor calibrations (~0.4 °C). Note that this comparison is between temperatures observed at slightly different heights; for context,
the average difference between the temperature observed at 15 m and 3 m from the 93 radiosondes was similar (-0.1 °C ± 0.14 °C).

### 3.2.3 Relative Humidity

Atmospheric relative humidity (RH) was also measured at 1 Hz by the IMET Rotronic MP-101A on the bow mast at approximately 15 m ASL, and by Rotronic MP-101A probes mounted on the TAO buoys at 3 m ASL. We compared the ship
RH and buoy RH for the same cases and using the same methodology as for temperature, with a different outcome (Fig. 15). There is a systematic bias of -5.7% ± 1.6% in the ship's RH observation compared to those from the buoys.  Also plotted in





Figure 15b is a comparison between the ship RH and the humidity reported by radiosondes while they were outside on the fantail (~4 m ASL) prior to launch. The comparison was only made for radiosondes launched during night (solar zenith angle > 90°) because of solar heating of the fantail deck during the day (Hartten et al., 2017a) and uses radiosonde data averaged over the 1 minute just prior to launch. This comparison was again between two different heights, ~ 15 m and ~ 4 m ASL.

Edson et al. (2004) documented strong near-surface vertical humidity gradients at approximately (3° S, 122° W to 150° W) during February 2001, using measurements made from the NOAA Ship *Ronald H. Brown* during GasEx-01, so we expect to see a difference of about 1.5%. The actual difference between our nighttime radiosonde RH profiles at these heights (15 m minus 4 m) is -1.6% ± 1.5%, consistent with this expectation. The ship-to-radiosonde comparison indicates that the ship RH is biased low (-4.0% ± 1.4%). Note that sensor response times and mixing of the ambient air by the superstructure of the ship

gives reason to exercise caution with respect to these results. The results nevertheless are consistent with the interpretation of the observed bias being instrumental error that is not explained by differences in height. We have therefore corrected the ship's recorded RH by adding a constant 4.0% throughout the cruise, resulting in an RH for an effective height of approximately 3 m. We chose to use the correction based on the comparison to the radiosondes instead of the buoys because the sample size is larger (n=99) and covers a longer range of time, and because the distance between the sensors was always

less 100 m.

### 3.2.4 Winds

Wind speed and direction during the cruise were measured by sonic anemometers mounted on poles on both the port and starboard sides of the bridge roof, roughly 19 m ASL. Three issues affected the measurements, one particular to this cruise and the others more general: one of the bridge anemometers was misaligned; flow is distorted when an anemometer is in the

20 lee of local obstacles; and the ship itself causes distortion of the wind flow at all points. Our corrections were complicated by the fact that the bow mast's anemometer was not operating and the ship was never co-located with a buoy that was collecting winds, both of which deprived us of independent measurements. The procedure we used to obtain what we consider to be a credible time series of wind speed and direction is described below.

We began by addressing the misaligned anemometer. After the field campaign was over we investigated this bias by examining the distribution of the differences in the wind direction when the wind speeds are in agreement and greater than 10 m s$^{-1}$ while the *ship-relative* wind direction recorded by the port sonic was between 320° and 360°. The latter condition ensures that winds are from the port-side bow, since initial analysis indicated possible obstruction when winds were from 0° to 45°. The results indicate that the starboard sonic was misaligned -17.6° *relative to the port sonic measurement*, which is assumed to be properly

aligned parallel to the ship's beam with "north" pointing in the direction of the bow. Therefore, all starboard sonic wind directions have been rotated clockwise by 17.6°, and all further discussion of starboard winds is based on the rotated values.



We next combined data from the two bridge anemometers into a single time series by selecting the windward anemometer at all times. Specifically, we used the starboard anemometer when it measured ship-relative winds between 0° and 170° and the port anemometer when it measured ship-relative winds between 190° and 360°. When both were within range, the starboard anemometer's winds were used because our analysis of the data indicates there is less localized distortion near the starboard

anemometer. When ship-relative winds were from the stern, which was quite rare, they were treated as missing. The resulting time series for the entire cruise is shown in Figure 16.

The problems associated with free-stream velocity issues were addressed in multiple steps. The typical procedure would be to use the winds measured on the bow mast and correct for flow distortion at that position, which is well-characterized for this

vessel. Moat et al. (2001) used computational fluid dynamics simulations to calculate the biases in measured wind velocities due to flow distortions caused by the NOAA Ship *Ronald H. Brown*'s superstructure. They made calculations for sensor positions on the bow mast, but not for the anemometer positions on the bridge roof. However even in the absence of calculations for the bridge positions, correction scale factors for an effective height in free-stream for the sonic anemometers on the bridge could be estimated from a comparison to the corrected bow-mast wind measurements.

Since we had no bow-mast winds during ENRR, we instead estimated correction factors from data acquired by the *Ronald H. Brown* in January and February 2015 during the CalWater-2 field campaign. That cruise occurred in the Pacific, northeast of Hawaii (Ralph et al., 2016), and data from both bridge-mounted sonic anemometers and from a sonic anemometer mounted on the bow mast were recorded. We assume that the bias in the bow mast's sonic is -5% in $v$ (parallel to ship's orientation)

and +15% in $u$ (transverse to ship's orientation) (C. Fairall and B. Blomquist, 2017, personal communication) and incorporate correction factors accordingly for the CalWater-2 data. The biases in $u$ and $v$ for both of the bridge sonic anemometers are shown in Figure 17 as functions of measured $u$ and $v$ at the bridge. These relationships are linear and so a single scale factor in $u$ and $v$, analogous to the corrections applied to the bow mast data, can be used to correct the bridge data. The correction factors were determined using the mean proportional error (vertical lines in Figure 18): -6.1% (Port $v$), +1.1% (Port $u$), -2.7%

(Starboard $v$), -7.1% (Starboard $u$). These corrections effectively convert data from the bridge sonic anemometers to match the corrected winds measured by the sonic mounted on the bow, which is at a different height and is in a different position on the ship. Since the bow sonic anemometer is positioned at 17.251 m, this becomes the effective height for the bridge sonic anemometer's data. We applied these correction factors to the wind data collected during ENRR at the bridge, then scaled them assuming neutral stability conditions using the wind profile power law with an exponent of 0.11 (Hsu et al., 1994) in

order to represent winds at the 3.8 m ASL height of the ground station.



### 3.2.5 Sea Surface Temperature, Downwelling Shortwave Radiation, and Ship Position and Movement

SSTs were logged at 1 Hz from a Sea Bird SBE45 thermosalinograph operating at a depth of 5 m. Downwelling shortwave radiation was logged at 1 Hz from an Eppley PSP. Latitude and longitude, as well as the ship's course and speed over ground, were recorded by a Furuno GP90, while ship heading was measured by a Sperry MK37, all at 1 Hz. These collected data were
5 averaged to 1 minute; no other changes were made.

### 4 Surface Fluxes at the NOAA Ship *Ronald H. Brown*

Based on the ENRR surface observations described in the previous section, we calculated bulk air-sea fluxes (1-hour averages) using a recent version of the Tropical Ocean and Global Atmosphere (TOGA) Coupled Ocean–Atmosphere Response Experiment (COARE) flux algorithm (Fairall et al., 1996). A complete list of the input data and the variables calculated from
10 it by the COARE flux algorithm are in Table 5. The full COARE algorithm requires some atmosphere/ocean quantities that were not measured during the cruises. With the exception of longwave radiation (LWD) we have not attempted to estimate these, so quantities that rely on them, such as rain rate and turbulent fluxes from Eddy Covariance methodology, have not been calculated. Longwave downwelling radiation (LWD) is needed for the bulk flux calculations within the COARE algorithm if a cool-skin correction is applied to SST; it is required if a full surface heat budget is to be calculated from the COARE results.
Unfortunately, LWD was not measured during this cruise and so we estimated it using other means, as described in 4.2.

### 4.1 Treatment of Measured Properties

We constructed 1-hour average time series of several quantities needed as input by the COARE algorithm. These are all computed from the one-minute surface data set described in Sect. 3.2: atmospheric pressure (3.8 m ASL); temperature (15 m); relative humidity (effective height 3 m); wind speed (valid at ~4 m ASL); ocean temperature (aka SST, valid at 5 m depth);
downwelling shortwave radiation; latitude and longitude.

### 4.2 Estimation of Downwelling Longwave Radiation

The LWD includes contributions from atmospheric gases and from clouds, and following convention we define those contributions as $LWD_{clr}$, the "clear-sky" flux from atmospheric gases only, and longwave cloud radiative effect (LWCRE), the enhancement to the LWD caused by clouds (e.g., McFarlane et al., 2013):

$$LWD = LWD_{clr} + LWCRE \qquad (1)$$

To determine LWD, we estimated $LWD_{clr}$ and LWCRE from observations we did collect plus a number of simple assumptions, as described below. The values reported are referenced to the height of the radiosonde ground station, 3.8 m ASL.



We first used the longwave Rapid Radiative Transfer Model (RRTM, version 3.3; Mlawer et al., 1997) to calculate $LWD_{clr}$ from each atmospheric sounding collected during the cruise. At each sounding time we provided the model with the "Level 2" radiosonde temperature profile (Cox et al., 2017a; Hartten et al., 2017b) as well as with profiles of the seven most significant radiatively active gases: $H_2O$, $CO_2$, $O_3$, $N_2O$, $CO$, $CH_4$ and $O_2$. The profile of $H_2O$ came from the same radiosonde profile as

5   the temperature. The $CO_2$ was set to 398.4 ppm, which was the in situ monthly mean surface value measured by NOAA/ESRL's Global Monitoring Division (GMD) at American Samoa in March 2015 (Thoning et al., 2015), and distributed using a constant mixing ratio at all heights. Profiles of the other gases were taken from the US Standard Atmosphere tropical profile (McClatchey et al., 1972). All data were interpolated to a pre-determined height grid consisting of 58 levels from 0 to 60 km (see Table 6). All levels above the termination of the radiosonde were filled using the standard atmosphere.

For each sounding we also calculated LWD using the measured temperature/humidity profile but for a scene containing a hypothetical optically-thick cloud positioned at 1 km. We used RRTM coupled with the Discrete Ordinate Radiative Transfer (DISORT; Stamnes et al., 1988), and considered only absorption, no scattering. The result is a rough estimate of the maximum LWD for that profile, i.e. the LWD in the presence of a low, optically-thick stratiform cloud. We were thus able to calculate a

maximum estimate of LWCRE, $LWCRE_{max}$ using Eq. (1).

Cloud optical depth was not measured during the cruise, but sky cover (in oktas) is available from hourly observations made from the bridge. We converted oktas to fractional sky cover by dividing by eight, linearly interpolated to the one-minute resolution of the surface observations, then multiplied the cloud fraction at each sounding time by the corresponding

$LWCRE_{max}$ to yield an estimate of the actual LWCRE. This LWCRE accounts for cloud fraction, including scattered cloud cover, but assumes all clouds are optically-thick; this assumption is reasonable since most of the clouds encountered during the cruise were low- to mid-level cumulus and cumulonimbus. Our final estimate of LWD at each sounding time was computed from Eq. (1) using the estimates of $LWD_{clr}$ and LWCRE. This was linearly interpolated to the one-minute resolution of the surface observations and averaged over each hour, and the resulting time series (Fig. 19) was then used as input to the COARE

flux algorithm. Since $LWD_{clr}$ is calculated directly from the observations, it is expected to be much more robust than the estimate of LWCRE, which requires more assumptions. From the figure, it is apparent that the $LWCRE_{max}$ was somewhat larger at the beginning and ending of the cruise when the total column water vapour was lower (not shown) (see also Cox et al. 2015) and fairly consistent during the middle part of the cruise when the ship was between 8° N and 8° S (the mean $LWCRE_{max}$ was 38.1 W m$^{-2}$ during this time). Since it was generally cloudy during the cruise (fractional cloud cover = 69%),

the mean LWCRE is estimated to be 31.1 W m$^{-2}$ overall, representing just 7.3% ±3.8% (1σ) of the LWD.




### 4.3 Bulk Flux Calculations and Final Data Set

We used the pressure, air temperature, wind, bulk ocean temperature, solar radiation, and longwave radiation time series described above as input to the COARE flux algorithm (version coare35vnWarm; Edson et al., 2013). We assumed a surface emissivity of 0.97 and a surface albedo of 0.945. Because we had a bulk ocean temperature rather than a true ocean skin

temperature, we had the algorithm do a cool-skin calculation. The resulting time series together with those surface time series used as input constitute the released bulk flux data set (Table 5). Figure 20a shows the net surface heat flux $Q_0$ calculated from some of the measurements and calculated quantities as

$$Q_0 = Q_{sw} + Q_{lw} - Q_{lat} - Q_{sen} , \tag{2}$$

where $Q_{sw}$ is the net (absorbed) solar irradiance, $Q_{lw}$ is the net longwave flux, $Q_{lat}$ is the bulk latent heat flux out of the ocean

and $Q_{sen}$ is the bulk sensible heat flux out of the ocean. For reference, both the observed air temperature and ocean temperature are plotted in Fig. 20b. The large amplitude of the diurnal cycle (500-1000 $Wm^{-2}$) indicates that uncertainty associated with the estimate of LWD is quite small relative to the magnitude of the other terms except for the times of the day when the net surface heat budget is switching from net warming to net cooling, and *vice versa* (near sunrise and sunset).

### 5 Data Availability and Use

The data sets described here are archived at NOAA's National Center for Environmental Information (NCEI), with DOIs as follows: surface meteorology from Kiritimati (doi:10.7289/V51Z42H4), surface meteorology and SST from the NOAA Ship *Ronald H. Brown* (doi:10.7289/V5SF2T80), and surface fluxes from the NOAA Ship *Ronald H. Brown* (NCEI Accession Number 0167875, available at http://accession.nodc.noaa.gov/0167875). They are also available, together with sample code to read the data, from NOAA/ESRL/PSD at https://www.esrl.noaa.gov/psd/enso/rapid_response/data_pub/. Users of these

data must cite the appropriate DOI and reference the data as indicated below:

Cox, C., and Hartten, L.: El Niño Rapid Response (ENRR) Field Campaign: Surface Fluxes from the NOAA Ship Ronald H. Brown, 2016-02 to 2016-03 (NCEI Accession 0167875), NOAA /National Centers for Environmental Information, doi pending, 2017.

Cox, C., Wolfe, D., Hartten, L., and Johnston, P.: El Niño Rapid Response (ENRR) Field Campaign: Surface Meteorological and Ship Data from the NOAA Ship Ronald H. Brown, February-March 2016, Version 1.1, NOAA /National Centers for Environmental Information, doi:10.7289/V5SF2T80, 2017.

Hartten, L., Johnston, P., Cox, C., and Wolfe, D.: El Niño Rapid Response (ENRR) Field Campaign: Surface Meteorological
Data from Kiritimati Island, January-March 2016, Version 1.1, NOAA /National Centers for Environmental Information, doi:10.7289/V51Z42H4, 2017.

The data from Kiritimati start on 25 January 2016 03:07:43 UTC and end on 28 March 2016 19:14:43 UTC, with four gaps in one or more parameters lasting an hour or more (Table 3). The ship data start on 16 February 2016 18:32 UTC and end on 16

March 2016 from 18:50–18:57 UTC, depending on the parameter.




Each data set is in the form of a single file written in the NASA Ames Format for Data Exchange (hereafter "NASA-Ames Format"); see Gaines and Hipskind (1998) as well as the British Atmospheric Data Centre (BADC) explanatory material (British Atmospheric Data Centre, 2002). NASA-Ames Format's plain-text (ascii) nature makes it portable to any machine

and easily accessible to users with limited computer resources; the rich metadata in the mandated and optional header sections make it self-describing. NASA-Ames Format requires that the total number of header lines be the first number in the first line of the file, and that the data following the header lines be space-delimited. Our files use the last header line to provide rudimentary column headers for the data that follow, which makes importing the data into spreadsheets and starting to work with them fairly straightforward. Time is the independent variable in all the files, and is presented as days since 1900-01-01

00:00:00 +00:00, i.e. since 1 January 1900 00:00:00 UTC. Users working with the data from Kiritimati Island, especially those interacting with residents and local records, should keep in mind that Kiritimati is in the Line Islands Time (LINT) time zone, UTC+14.

## 6 Conclusions

As the 2015/2016 El Niño gathered strength, NOAA's ESRL/Physical Sciences Division conceived the idea of a rapidly

deployed multi-platform field campaign to meet a variety of operational and research goals (Dole et al., 2017). Observations during the ENRR field campaign, which involved many partners in and outside of NOAA, included almost 10 weeks of surface meteorology from Kiritimati Island and four weeks of surface meteorology from the NOAA Ship *Ronald H. Brown* in the central equatorial Pacific. These data, which were collected at both sites primarily to support radiosonde measurements, have issues caused primarily by instrument failure, out-of-calibration instruments, and less than optimal instrument placement.

Some of the issues are a side-effect of the rapid nature of the campaign planning and deployment; others are endemic on crowded research vessels or in remote and geographically challenging locations.

We have carefully vetted the data against expectations and alternate observations when available, identifying issues and minimizing their impacts by making corrections when possible. We have also estimated air-sea surface fluxes from the ship's

data. The final data products that are described in this paper have been placed in a long-term repository and are freely accessible for users. Each file contains considerable internal documentation; this article serves as a detailed description of the instrumentation, siting, issues, and corrections. Highlights of the data sets include the high-resolution measurements of nearly a meter of rainfall at Kiritimati and continuous measurements of ocean and air temperatures in the heart of the El Niño warming as the ship visited the TAO buoys along 140° W and 125° W. These data represent a rich resource for local studies, for

inclusion in models of local/regional processes, or for comparison with satellite observations or model simulations.

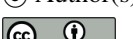



## Appendix A: Calculation details for circular statistics

The circular statistics presented in Table 4 and discussed in Section 3.1.3 are predominantly based on Fisher (1995) and MATLAB code described in Berens (2009), itself strongly reliant on Fisher's book. Since most of the statistics are not yet often used in meteorological work, and since some alterations have been made to Berens' code, the calculations are described

here. Discussion of the rationale and details behind the equations is to be found in the cited references.

The sample mean direction $\bar{\theta}$ and mean resultant length $\bar{R}$ are obtained by vector averaging the $n$ winds in the sample (Fisher 1995, Eq. (2.7–2.9)). The sample median $\tilde{\theta}$ is found by determining "the diameter that divides the data into two equal sized groups", and is either the wind direction nearest "the endpoint of the diameter closer to the center of mass of the data" (for an

odd number of wind values) or "half-way between the two closest" wind directions (for an even number of wind values) (Berens, 2009). The sample circular dispersion $\hat{\delta}$, one measure of the spread of the data, is calculated as per an online Erratum (Fisher, 2017) using

$$\hat{\delta} = (1 - \hat{\alpha}_2)/(2\bar{R}^2) \tag{A1}$$

where

$$\hat{\alpha}_2 = (1/n) \sum_{i=1}^{n} \cos 2(\theta_i - \bar{\theta}). \tag{A2}$$

This calculation departs from Eq. (2.28) in Fisher (1995),

$$\hat{\delta} = (1 - \hat{\rho}_2)/(2\bar{R}^2), \tag{A3}$$

where

$$\hat{\rho}_p = \sqrt{[(1/n) \sum_{i=1}^{n} \cos p(\theta_i - \bar{\theta})]^2 + [(1/n) \sum_{i=1}^{n} \sin p(\theta_i - \bar{\theta})]^2} \tag{A4}$$

(Fisher, 1995, Eq. (2.25); also Berens, 2009). However, the Erratum's amendment to this form given above makes sense given Fisher's definition of $\hat{\rho}_2$ as

$$\hat{\rho}_2 = (1/n) \sum_{i=1}^{n} \cos 2(\theta_i - \bar{\theta}) \tag{A5}$$

in his Eq. (2.27).

In a normal distribution, the dispersion about the mean is discussed in terms of the variance; in circular statistics the dispersion of the wind about the mean direction is discussed in terms of the sample concentration parameter $\hat{\kappa}$. This can be simulated with some difficulty (Fisher, 1995, Section 3.3.6) or else estimated as a function of the mean resultant length $\bar{R}$ using a table (e.g. Fisher, 1995, Appendix 3) or a conditional formula (Fisher, 1995, Eq. (4.40)):

$$\hat{\kappa} = \begin{cases} 2\bar{R} + \bar{R}^3 + \frac{5\bar{R}^5}{6}, & \bar{R} < 0.53 \\ -0.4 + 1.39R + \frac{0.43}{(1-\bar{R})}, & 0.53 \leq \bar{R} < 0.85 \\ \frac{1}{(\bar{R}^3 - 4\bar{R}^2 + 3\bar{R})}, & \bar{R} \geq 0.85 \end{cases} \tag{A6}$$

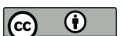



The CircStat MATLAB code (Berens, 2009), which employs the latter, was used to obtain the values in Table 4.

Finally, Fisher (1995, Section 5.3.4) outlines a method of testing whether two or more samples have a common mean direction. Since both our samples consist of more than 25 observations and their dispersions are considered comparable because $\frac{\hat{\delta}_{max}}{\hat{\delta}_{min}} \leq$

4, we used Fisher (1995, Eq. (5.10–5.13)) to calculate the test statistic

$$Y_2 = 2(N - R_P)/\hat{\delta}_0 \tag{A7}$$

where N is the sum of the two sample sizes,

$$R_P = \sqrt{[\sum_{i=1}^{2} n_i \cos \bar{\theta}_i]^2 + [\sum_{i=1}^{2} n_i \sin \bar{\theta}_i]^2} , \tag{A8}$$

and

$\hat{\delta}_0 = \sum_{i=1}^{2} n_i \hat{\delta}_i/N$ . $\tag{A9}$

**Author Contributions.** All the authors spent at least two weeks at Kiritimati or four weeks aboard the NOAA Ship *Ronald H. Brown* collecting data. S. Abbott set up and later relocated the Kiritimati surface instruments, P. E. Johnston modified the software for the Kiritimati datalogger to give 1 minute data and higher resolution pressure values, and H. A. McColl replaced

the power supply after the solar panel failed. P. E. Johnston corrected the surface data from Kiritimati with input from L. M. Hartten; C. J. Cox corrected the surface data from the NOAA Ship *Ronald H. Brown* with input from D. E. Wolfe. C. J. Cox generated the surface flux data from the NOAA Ship *Ronald H. Brown*. L. M. Hartten assembled metadata and created the file headers with input from all C. J. Cox, P. E. Johnston, and D. E. Wolfe, then created the final files from P. E. Johnston's and C. J. Cox's corrected surface meteorological data and C. J. Cox's surface flux data. L. M. Hartten prepared the manuscript

with input from all co-authors.

**Competing Interests.** All authors declare that they have no conflict of interest.

**Acknowledgements.** The ENRR Field Campaign was funded by NOAA through the Office of Oceanic and Atmospheric

Research (OAR) as well as the National Weather Service (NWS), the Office of Marine and Aviation Operations (OMAO) and the National Environmental Satellite, Data and Information Service (NESDIS). We thank the Republic of Kiribati for allowing NOAA to conduct research on Kiritimati Island, and the Kiribati Meteorological Service for their interactions with our observers during and after the field campaign. We appreciate the support we received from the crew of the NOAA Ship *Ronald H. Brown*, in particular NOAA/OMAO Survey Technicians J. Gunter and M. Bradley and NOAA/OMAO Lieutenant A.

Hopper. The authors were joined in collecting these data by X.-W. Quan (CIRES and ESRL/PSD) on Kiritimati Island and by M. G. Winterkorn (NVision Solutions and NOAA/ National Data Buoy Center) on the NOAA Ship *Ronald H. Brown*. The Kiritimati-based team is grateful to the management and staff of the Captain Cook Hotel, who welcomed ENRR observers and



equipment into their lives and went out of their way to make our stay very pleasant, and to J. Bryden (JMB Enterprises), whose logistical and supply support was critical to our success. Thanks to C. Fairall (ESRL/PSD), B. Blomquist (CIRES and ESRL/PSD), and P. M. Rowe (Northwest Research Associates) for consultations regarding producing surface flux data, C. A. Smith (CIRES and ESRL/PSD) for helping to navigate archival waters, D. Hooper (CIRES and ESRL/PSD) for scripting

magic, and C. Fairall and R. Zamora (ESRL/PSD) for carefully reading the manuscript. The colours used in many figures are from sets at ColorBrewer.org.

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

10   **its north indicating the initial and final locations of the surface meteorological instruments.**



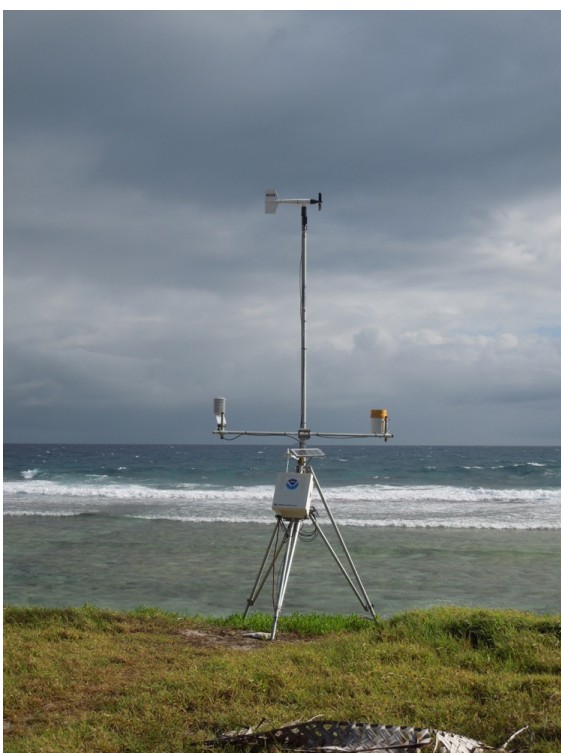

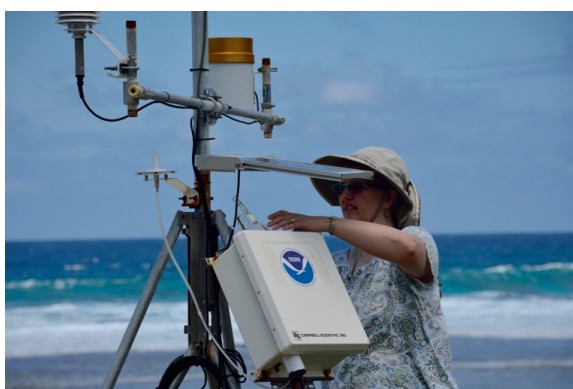

**Figure 2: (a) The surface met station in its final position above the beach at Kiritimati. The anemometer sits atop the assembly, while the temperature/humidity sensor (left) and rain gauge (right) are attached to the crossbeam. Just below the crossbeam is the**
5 **solar panel, and the box containing the Campbell datalogger, the power supply, and the barometer is mounted just below that. A plastic tube runs from the barometer out the tube on the bottom of the box, and can be seen mounted within a white disk just to the left of the solar panel. (b) The 26 March 2017 00 UTC radiosonde being put on the Campbell box in the shade of the solar panel prior to launch. The pressure tube egress and mount are clearly visible.**





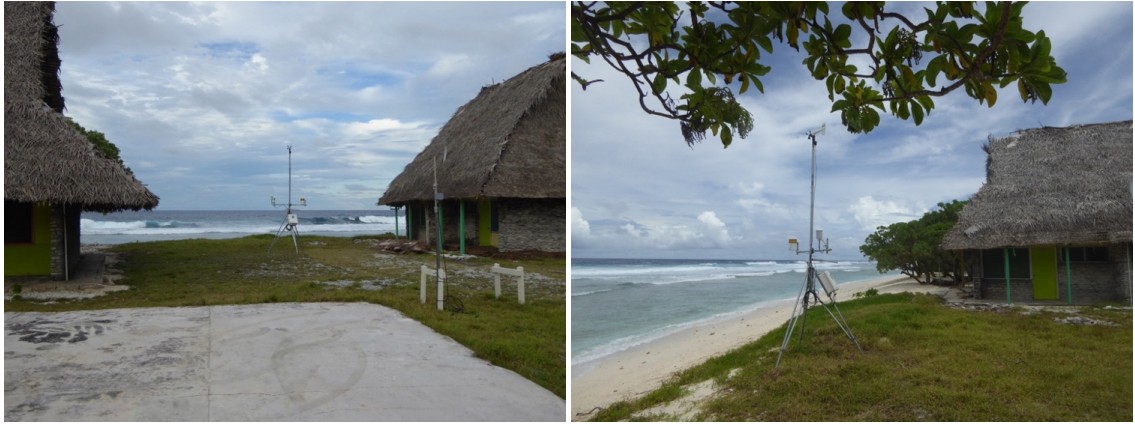

**Figure 3: The original position of the surface instruments (left), and their location after being moved back towards the top of the beach on 5 Feb 2017 (right).**

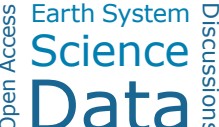

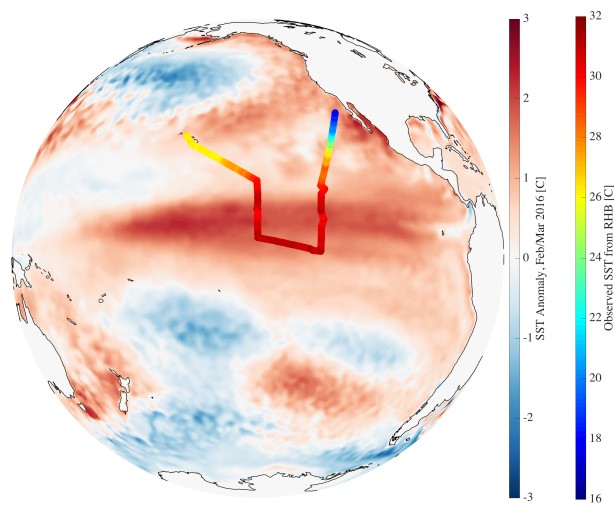

**Figure 4: SSTs observed by the NOAA Ship *Ronald H. Brown* during ENRR, overlaid on SST anomalies. The latter are the departure from the ERA-Interim (Dee et al., 2011) 1979-2016 February through March mean.**



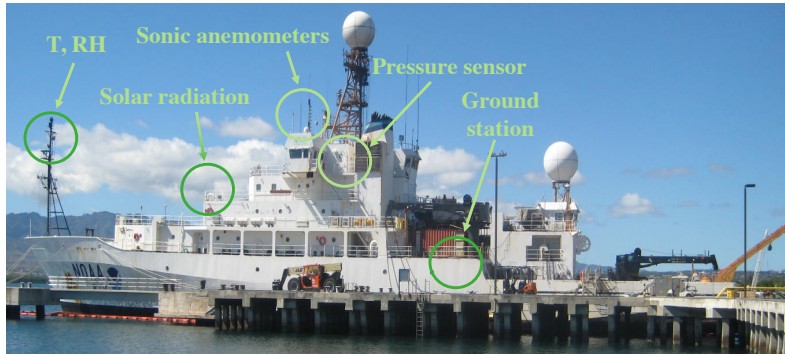

**Figure 5: The location of key surface meteorological instruments on the NOAA Ship *Ronald H. Brown*.**





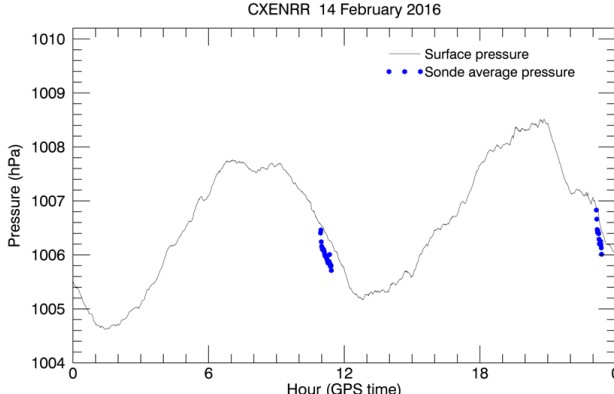

**Figure 6: Surface pressure measured by the PTB101B (1-minute resolution) on 14 February 2016 together with pressures measured by the day's two radiosondes prior to launch (1-minute averages). The first two points of the averaged radiosonde data are omitted; one is a partial average beginning just after the radiosonde was powered up on the battery, and the second was from inside the room rather than outdoors on the datalogger box.**



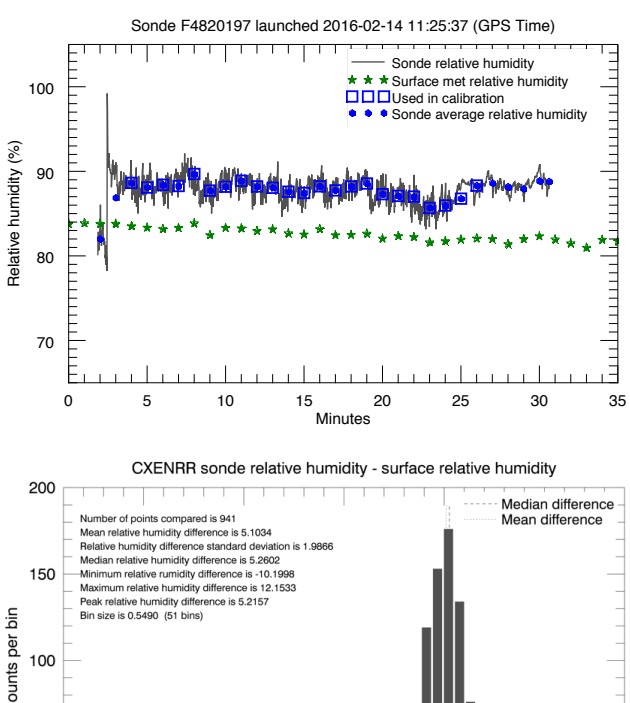

**Figure 7: a) One-second radiosonde relative humidity and one-minute average radiosonde relative humidity prior to the 14 February**
**2016 12 UTC launch, together with one-minute values of HMP45C relative humidity. The initial portion of the radiosonde data**
**cover the period when the radiosonde was transported from the ground check unit inside the bungalow out to the surface station.**
**The averaged values from this radiosonde that were part of the set used to calibrate the surface humidity instrument are indicated.**
**b) Differences between one-minute average radiosonde humidities and one-minute values of relative humidities from the HMP45C**
**when the radiosonde was co-located with the surface instruments prior to launch. All available good surface data and the**
**corresponding radiosonde data (101 radiosondes) are included.**



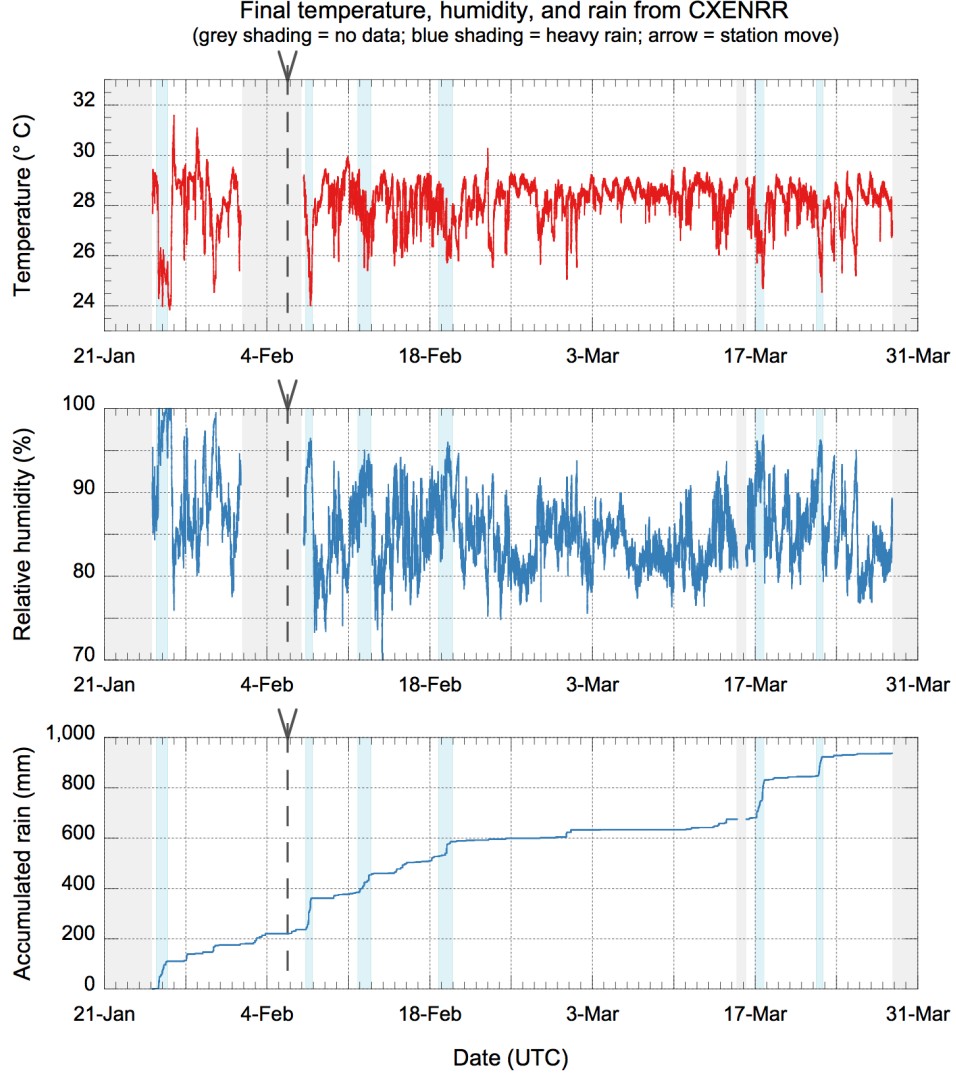

**Figure 8: The final time series of temperature (top) and relative humidity (middle) from Kiritimati Island, together with the accumulated rainfall (bottom) during the deployment. Pale blue shading indicates the six largest rainfall events, and is repeated on all graphs for reference.**

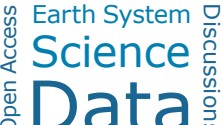



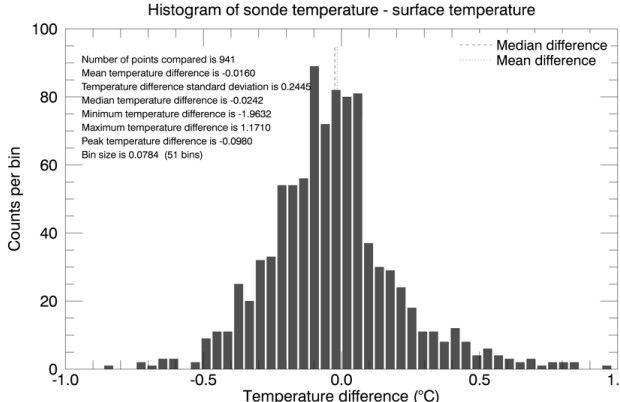

**Figure 9: Differences between one-minute average radiosonde temperatures and one-minute values of temperatures from the HMP45C when the radiosonde was co-located with the CXENRR surface instruments prior to launch. All available good surface data and the corresponding radiosonde data (101 radiosondes) are included.**




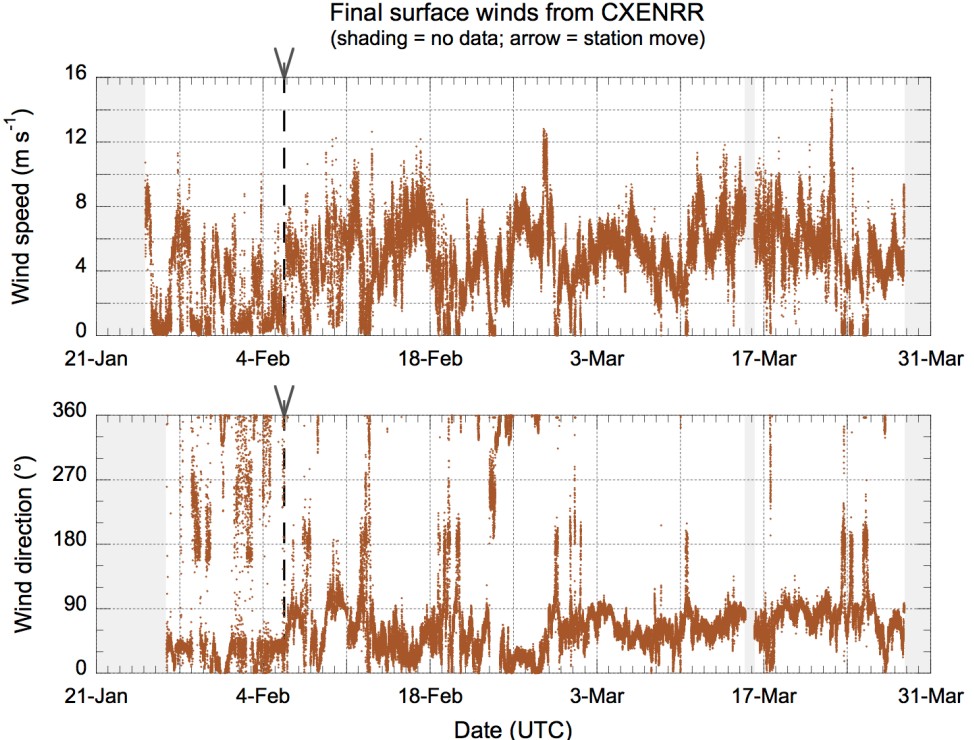

Figure 10: The corrected surface winds from the Kiritimati Island site.





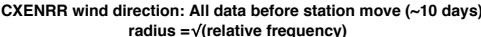

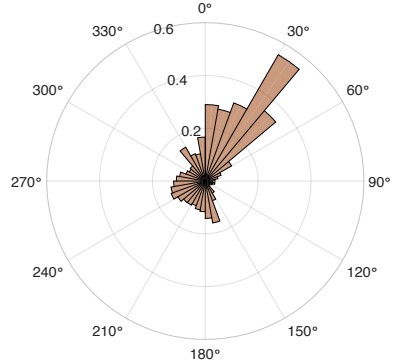 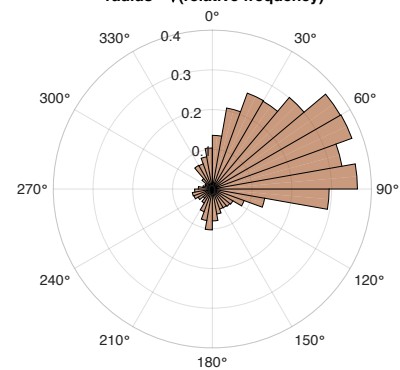

**Figure 11: Rose diagrams of all data collected before (left) and after (right) the CXENRR station was moved on 5 February 2016. Sector lengths are proportional to the square root of the relative frequency of that 10° bin, so the area of each sector is proportional to that bin's frequency. All data before the move are 2-minute averages; there are about 5.25 days of such data after the move, and thereafter the data are 1-minute averages.**



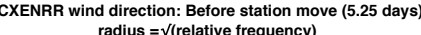

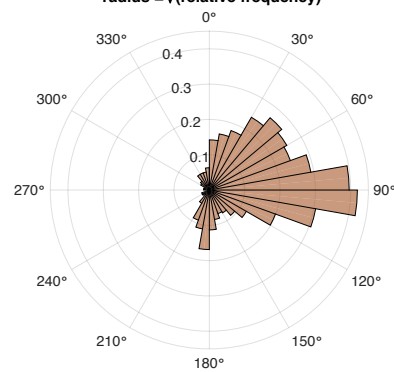

**Figure 12: Rose diagrams of 5.25 days' worth of 2-minute average collected immediately before (left) and after (right) the CXENRR station was moved on 5 February 2016. Sector lengths are proportional to the square root of the relative frequency of that 10° bin, so the area of each sector is proportional to that bin's frequency.**

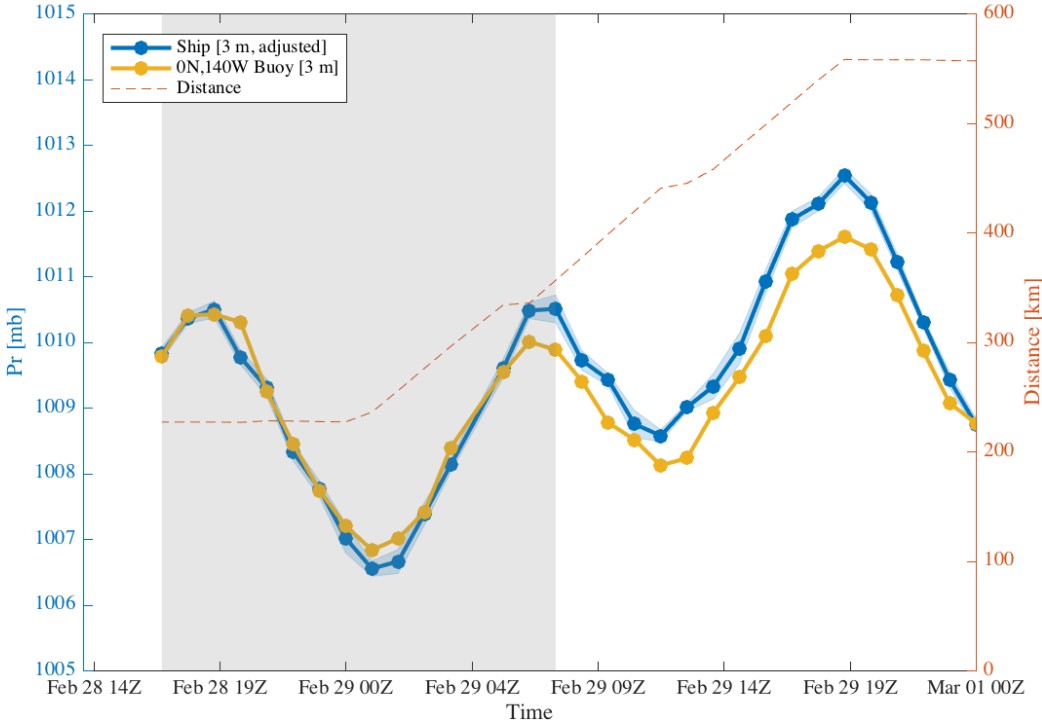

**Figure 13: Barometric pressure at 3 m ASL from the ship (blue) and from buoy dm166b (yellow), 28 February through 1 March 2016. Each observation is a 2-min average centered on the top of the hour. Blue shading shows the 1σ variability of the 2-min average from the ship; this information is not available from the buoy. The red-dashed line shows the distance between the ship and the buoy in km. For the time period when the ship was < 350 km from the buoy (gray shading), the difference (ship minus buoy) is -0.03 ± 0.3 hPa.**



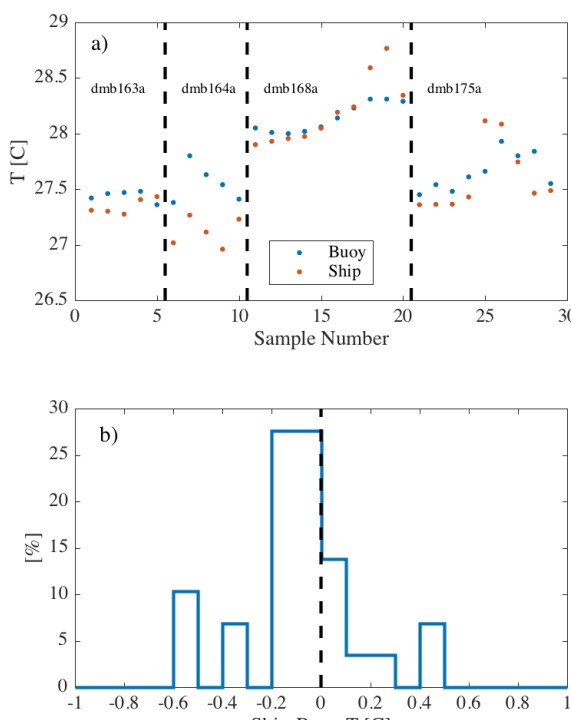

**Figure 14: Comparisons between the air temperatures measured by the ship and by a buoy when the ship was within 0.1°
latitude/longitude of the buoy. All data are 10 minute averages; ship measurements are from 15 m and buoy measurements from 3
m ASL.  a) All available observation pairs, with dashed vertical lines separating buoys: dmb163a (9° N, 140° W), dmb164a (5° N,
140° W), dmb168a (5° S, 140° W), and dmb175a (8° N, 125° W).  b) The distribution of all differences between those pairs.**



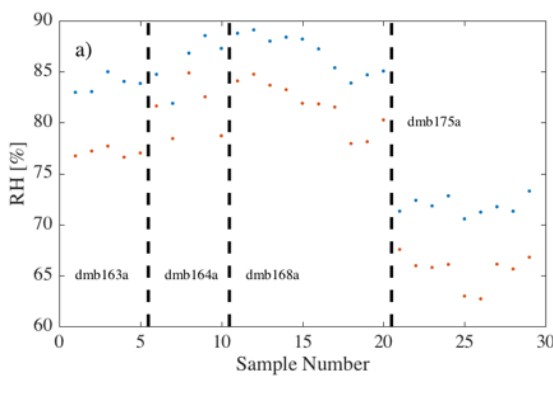

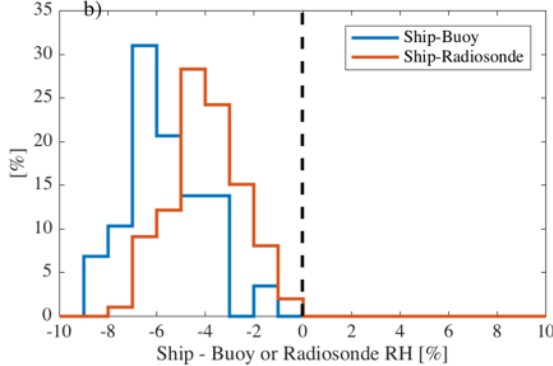

**Figure 15: Comparisons between the RH measured by the ship and by a buoy when the ship was within 0.1° latitude/longitude of the buoy. All data are 10 minute averages; ship measurements are from 15 m and buoy measurements from 3 m ASL. a) All**
5    **available observation pairs, with dashed vertical lines separating buoys: dmb163a (9° N, 140° W), dmb164a (5° N, 140° W), dmb168a (5° S, 140° W), and dmb175a (8° N, 125° W). b) The distribution of all differences between paired ship and buoy RHs, together with the differences between 99 pairs of ship RH and radiosonde RH, the latter from 4 m ASL.**

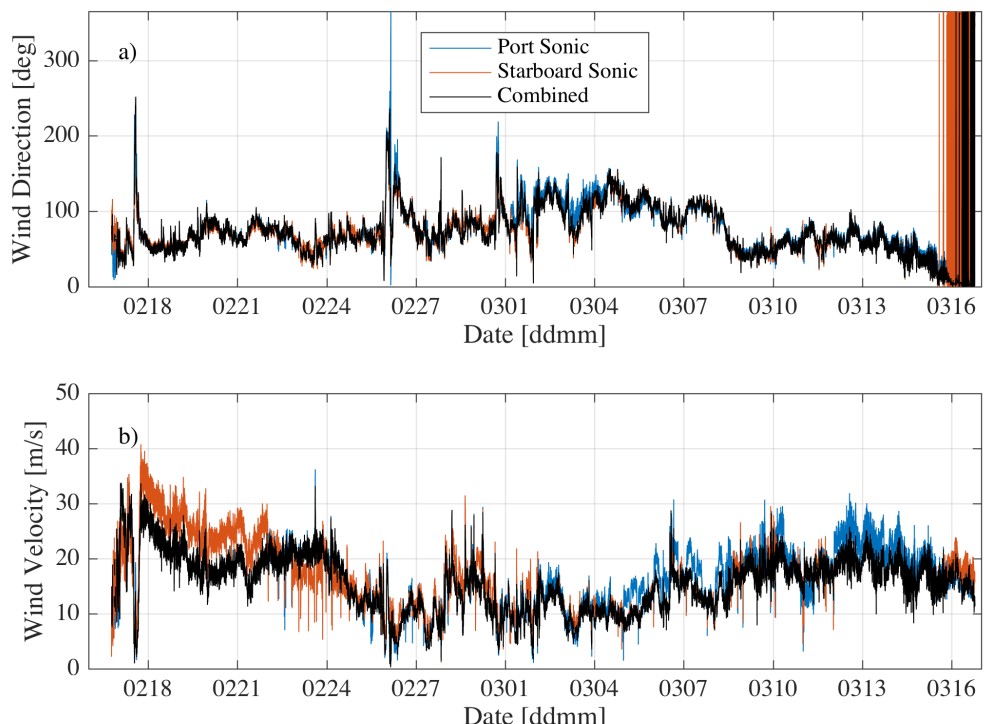

**Figure 16: a) Time series of wind directions during the entire cruise, as measured by the port side and starboard side ultrasonic anemometers and the time series constructed by combining the two. These winds have not been corrected for flow distortion. b)**
5 **Same as in (a), but for wind velocity.**



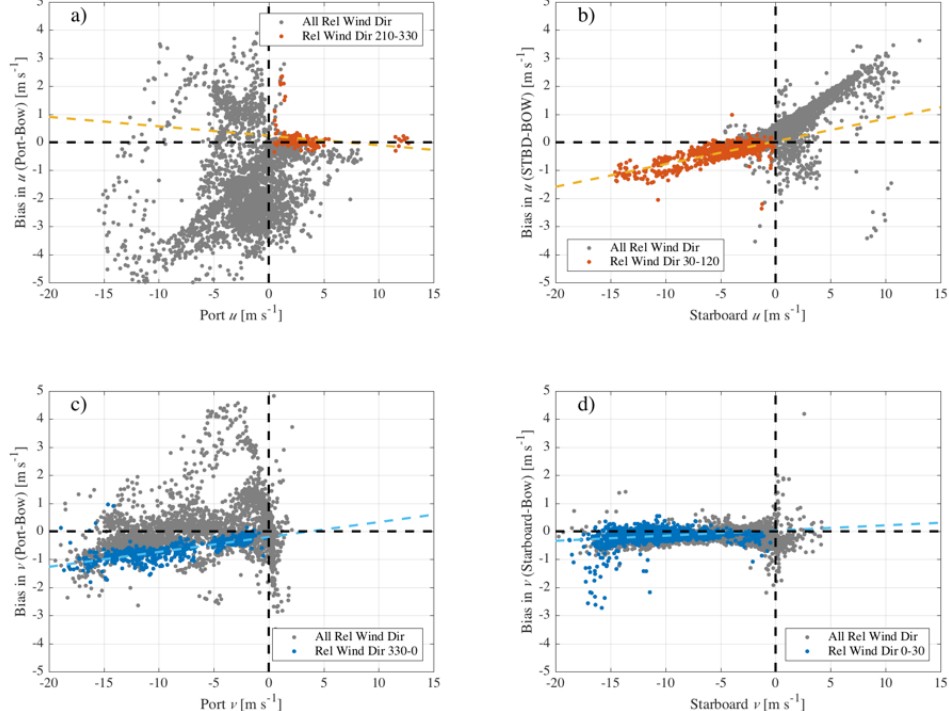

**Figure 17: Comparisons between winds measured by the bridge sonic anemometers ("Port" and "Starboard") and the bow mast anemometer ("Bow") on the NOAA Ship _Ronald H. Brown_ during CalWater-2, 2015. a) The bias in _u_ (Port – Bow) as a function of observed wind speed at Port, b) similar to (a), but for the Starboard sonic, c) and d) similar to (a) and (b) for the _v_ wind component. _u_ is defined perpendicular to ship's orientation and _v_ is defined parallel.**





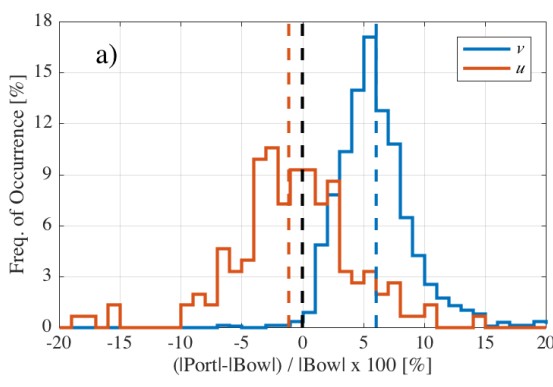

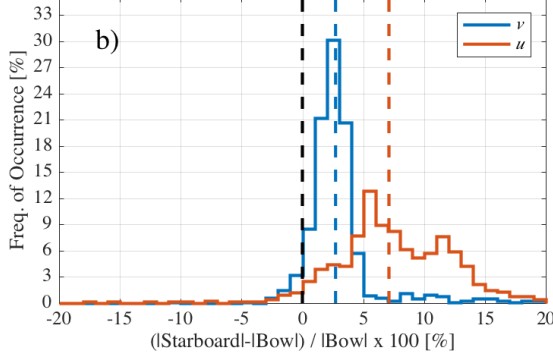

**Figure 18: Distributions of the percent error in the (a) port (a) and (b) starboard bridge sonic anemometers relative to the bow-mast sonic during CalWater-2. The data are the same as in Figure 14, with *u* defined as relative wind directions 210°-330° and 30°-120° and *v* defined as relative wind directions 330°-30°, but here are shown normalized by wind speed to illustrate the source of the correction factors (denoted by the coloured vertical-dashed lines) that are reported in the text.**





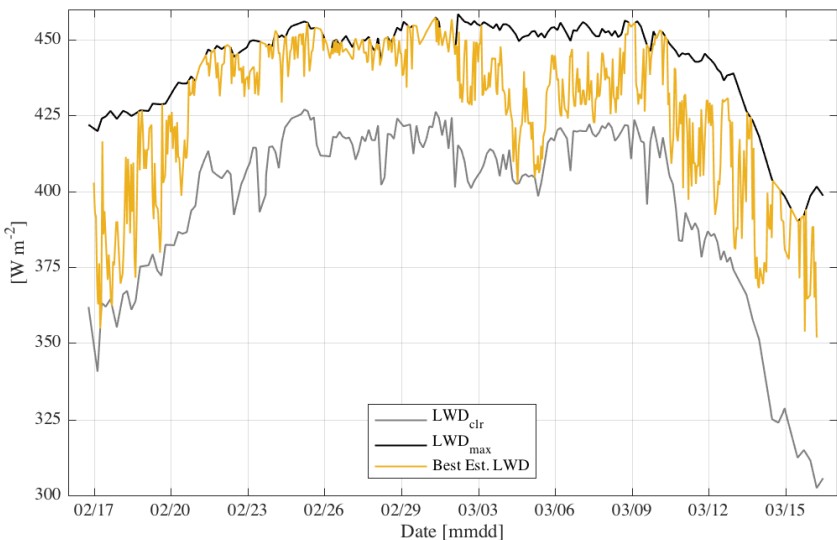

**Figure 19: Time series of our best estimate of hourly longwave downwelling radiation (LWD) during the ENRR cruise, compared with hourly estimates during clear-sky conditions, LWD$_{clr}$, and in the presence of a hypothetical optically thick cloud scaled by the observed cloud area fraction with a base height of 1 km ASL, LWD$_{max}$.**



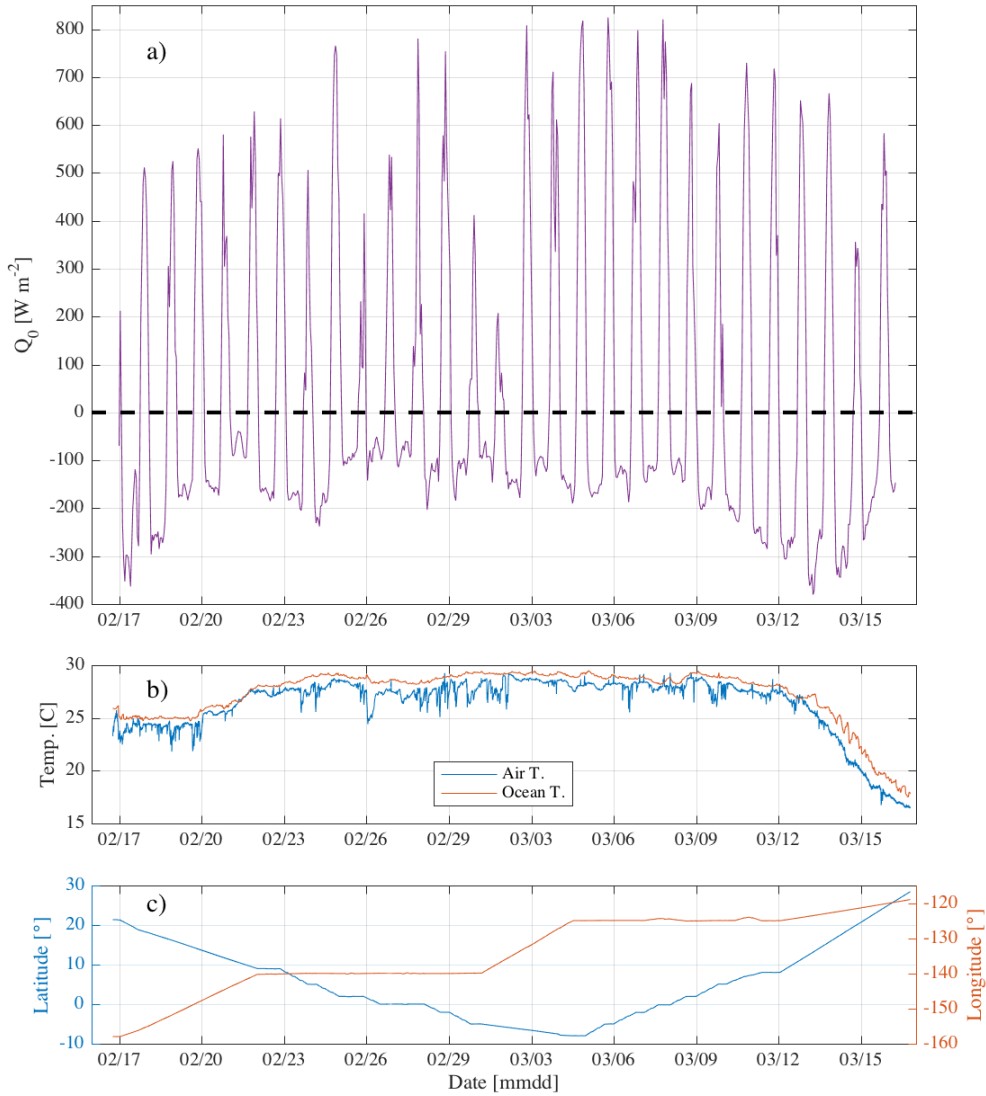

**Figure 20: a)** One-hour average ocean net surface heat flux ($Q_0$) along the NOAA Ship *Ronald H. Brown*'s track from 16 February–17 March 2016. The flux is based on a combination of measurements and calculations with the COARE bulk flux algorithm. **b)** For the same time period, one-minute average ocean temperature at 5 m depth and air temperature. **c)** Ship position.



**Tables**

TABLE 1. Surface meteorology instruments deployed at Kiritimati Island during the ENRR Field Campaign.

| Instrument | Height (ASL) | Parameter | Collection rate; Logged values | Accuracy | Operational Notes |
|---|---|---|---|---|---|
| Vaisala PTB101B attached to a SPH10 static pressure head | 3.6 m | Atmospheric pressure | 1 Hz; averaged[a] | ±0.5 hPa at 20 °C ±1.5 hPa at 0 °C to 40 °C | |
| Vaisala HMP45C in an R.M. Young 41003-5 solar radiation shield | 4.4 m | Air temperature | 1 Hz; averaged[a] | ±0.2 °C at 20 °C ±0.25 °C at 30 °C | Sensor bad starting 1 February 2016 17:59:43 UTC and was replaced 7 February 2016 03:41:43 UTC |
| | | Relative humidity | 1 Hz; averaged[a] | at 20 °C: ±2% (0–90% RH) ±3% (90–100% RH) | |
| R.M. Young 5103 anemometer | 6.6 m | Wind speed | 1 Hz; vector-averaged[a] | ±0.3 m s$^{-1}$ or 1% of reading | |
| | | Wind direction | | ±3° | |
| | | Maximum wind speed | 1 Hz; maximum during collection[a] | | |
| Texas Electronics TE525 tipping bucket | 4.4 m | Rain | Accumulation during collection[a] | ±1% up to 25.4 mm h$^{-1}$ +0, 3% from 25.4 to 50.8 mm h$^{-1}$ +0, −5% from 50.8 to 76.2 mm h$^{-1}$ | 0.254 mm per tip |
| Campbell Scientific CR23X datalogger | 3.6 m | Battery voltage | 1 Hz; averaged[a] | n/a | Power supply failed; no data logged from 15 March 2016 11:12:43 UTC to 16 March 2016 05:04:42 UTC |

[a] The averaging or collection period was 2 minutes through 2016-02-11 04:05:43. It changed to 1 minute starting 2016-02-11 04:06:43.





TABLE 2. Surface meteorology instruments deployed aboard the NOAA Ship *Ronald H. Brown* during the ENRR Field Campaign. The Improved Meteorology (IMET, Hosom et al. 1995) instruments are modified versions of the original manufacturer instruments, as described in ASIMET Sensors (2017) and ASIMET Module Specifications (2017).

| Instrument | Height (ASL) | Parameter | Collection rate; Logged values | Accuracy | Operational Notes |
|---|---|---|---|---|---|
| Vaisala PTB330 | 15.24 m | Atmospheric pressure | 1 Hz; averaged[a] | ±0.20 hPa at 20 °C ±0.25 (B) at 0 °C to 40 °C | Mounted on bridge Calibrated 29 December 2015 |
| IMET Rotronic MP101A | approx. 15 m | Air temperature | 1 Hz; averaged[a] | ±0.05 °C | Mounted on bow mast Calibrated 26 July 2014 |
| | | Relative humidity | | ±1% | |
| Vaisala WMT700 ultrasonic anemometers | approx. 19 m | Wind speed | 1 Hz; vector-averaged[a] | ±0.1 m s$^{-1}$ or 2% of reading | Mounted on poles atop the port and starboard sides of the bridge roof |
| | | Wind direction | | ±2° | |
| Sea Bird SBE45 thermosalinograph | -5 m | Sea surface temperature | 1 Hz; averaged[a] | ±0.002 °C | |
| IMET Eppley PSP | 11.2 m | Downwelling shortwave radiation | 1 Hz; averaged[a] | ±2 W m$^{-2}$ | |
| Furuno GP90 | approx. 17 m | Latitude, Longitude | 1 Hz; averaged[a] | ±10 m | |
| | | Course over ground | | ±3° at 1–17 kt ±1° at >17 kt | |
| | | Speed over ground | | 0.2 kt at ≤ 10kt 2% of ship speed | |
| Sperry MK37 | unknown | Heading | 1 Hz; averaged[a] | unknown | |

[a] The averaging or collection period was 1 minute.



TABLE 3. Major gaps in the Kiritimati surface data set. Data collection started 25 January 2016  03:07:43 UTC and ended 28 March 2016  19:14:42 UTC.

| Start of Gap (UTC) | End of Gap (UTC) | Reason | Measurement(s) Affected |
|---|---|---|---|
| 25 January 2016  03:07:43 | 26 January 2016  21:49:43 | Sensor not aligned | Wind direction |
| 1 February 2016  17:59:43 | 7 February 2016  03:41:43 | Sensor failure and replacement | Temperature, Relative Humidity |
| 5 February 2016  21:21:43 | 5 February 2016  22:01:43 | Surface station relocation | All |
| 15 March 2016  11:12:43 | 16 March 2016  05:04:42 | Solar panel failure and power supply replacement | All |



TABLE 4. Circular statistics computed from 5.25 days' worth of 2-minute average CXENRR winds collected immediately before (left) and after (right) the CXENRR station was moved on 5 February 2016. The 3° directional correction discussed in section 3.1.3 was applied before statistics were calculated. Statistical definitions are given in Appendix A1.

| Parameter | Before (31 January 2016 15:19:43 UTC – 5 February 2016 21:19:43 UTC) | After (5 February 2016 22:03:43 UTC – 11 February 2016 4:03:43 UTC) |
|---|---|---|
| Number of observations $n$ | 3780 | 3765 |
| Mean direction $\bar{\theta}$ | 18.8° | 81.1° |
| Mean resultant length $\bar{R}$ | 0.63 | 0.77 |
| Median direction $\tilde{\theta}$ | 302.0° | 86.6° |
| Circular dispersion $\hat{\delta}$ | 0.54 | 0.45 |
| Concentration Parameter $\hat{\kappa}$ | 1.65 | 2.56 |
| | **Summary** | |
| $\hat{\delta}_{max}/\hat{\delta}_{min}$ | 1.2 | |
| Test statistic $Y_2$ | 4262.4 | |



TABLE 5. Variables, all at hourly resolution, contained in the surface flux data file. The RRTMv3.3 algorithm is described in Mlawer et al. (1997) and the coare35vnWarm algorithm in Edson et al. (2013). The height of the input winds, 3.8 m ASL, is denoted by $z_u$ and the Obukhov length by $L$.

| Description | Source | Abbreviation | Units |
|---|---|---|---|
| *Select Input* | | | |
| Time | ship observations | Time(UTC) | days |
| Year, month, day, hour, minute, second | ship observations | Y, M, D, h, m, s | |
| Latitude | ship observations | Lat | ° N |
| Longitude | ship observations | Lon | ° E |
| Sky cover | ship observations | SCV | |
| Clear-sky longwave downward radiation | RRTMv3.3 | LWDclr | W m$^{-2}$ |
| *Surface Energy Budget* | | | |
| Longwave downward radiation at $z_u$ | calculated from LWD$_{clr}$, LWCRE | LWD | W m$^{-2}$ |
| Longwave upward radiation | coare35vnWarm | LWU | W m$^{-2}$ |
| Shortwave downward radiation | ship observations | SWD | W m$^{-2}$ |
| Shortwave upward radiation | coare35vnWarm | SWU | W m$^{-2}$ |
| Bulk latent heat flux out of the ocean | coare35vnWarm | hlb | W m$^{-2}$ |
| Bulk sensible heat flux out of the ocean | coare35vnWarm | hsb | W m$^{-2}$ |
| Bulk buoyancy flux into the ocean | coare35vnWarm | hbb | m$^2$ s$^{-3}$ |
| *Monin-Obhukov Similarity* | | | |
| Friction velocity that includes gustiness | coare35vnWarm | usr | m s$^{-1}$ |
| Wind stress | coare35vnWarm | tau | N m$^{-2}$ |
| Temperature scaling parameter | coare35vnWarm | tsr | K |
| Specific humidity scaling parameter | coare35vnWarm | qsr | g kg$^{-2}$ |
| Thermal roughness length | coare35vnWarm | z0t | m |
| Moisture roughness length | coare35vnWarm | z0q | m |
| Wind stress transfer (drag) coefficient at $z_u$ | coare35vnWarm | Cd | |
| Sensible heat transfer coefficient (Stanton number) at $z_u$ | coare35vnWarm | Ch | |
| Latent heat transfer coefficient (Dalton number) at $z_u$ | coare35vnWarm | Ce | |
| Obukhov length scale | coare35vnWarm | L | m |
| Monin-Obukhov stability parameter, $z_u/L$ | coare35vnWarm | zet | |
| *Surface Conditions* | | | |
| Wind speed adjusted to 10 m | coare35vnWarm | U10 | m s$^{-1}$ |
| Air temp at 10 m | coare35vnWarm | T10 | °C |
| Specific humidity at 10 m | coare35vnWarm | Q10 | g kg$^{-1}$ |



| | | | |
|---|---|---|---|
| Relative humidity at 10 m | coare35vnWarm | RH10 | % |
| Neutral value of wind speed at $z_u$ | coare35vnWarm | UN | m s$^{-1}$ |
| Neutral value of wind speed at 10 m | coare35vnWarm | UN10 | m s$^{-1}$ |
| Neutral value of drag coefficient at 10 m | coare35vnWarm | Cdn_10 | |
| Neutral value of Stanton number at 10 m | coare35vnWarm | Chn_10 | |
| Neutral value of Dalton number at 10 m | coare35vnWarm | Cen_10 | |
| Sea surface temperature | oare35vnWarm | SST | °C |
| Cool-skin temperature depression | coare35vnWarm | dter | °C |
| Surface saturation specific humidity | coare35vnWarm | Qs | g kg$^{-1}$ |
| Latent heat of vaporization | coare35vnWarm | Le | J kg$^{-1}$ |
| Evaporation rate | coare35vnWarm | Evap | mm h$^{-1}$ |

TABLE 6. The vertical grid to which all gas profiles ($H_2O$, $CO_2$, $O_3$, $N_2O$, $CO$, $CH_4$ and $O_2$) were interpolated before being used as input to the Rapid Radiative Transfer Model (RRTMv3.3).

| Heights (m) | | | |
|---|---|---|---|
| 0.0 | 1,000.0 | 5,000.0 | 20,000.0 |
| 3.8 | 1,200.0 | 5,500.0 | 25,000.0 |
| 10.0 | 1,400.0 | 6,000.0 | 30,000.0 |
| 25.0 | 1,600.0 | 6,500.0 | 35,000.0 |
| 50.0 | 1,800.0 | 7,000.0 | 40,000.0 |
| 75.0 | 2,000.0 | 7,500.0 | 50,000.0 |
| 100.0 | 2,250.0 | 8,000.0 | 60,000.0 |
| 150.0 | 2,500.0 | 9,000.0 | |
| 200.0 | 2,750.0 | 10,000.0 | |
| 250.0 | 3,000.0 | 11,000.0 | |
| 350.0 | 3,250.0 | 12,000.0 | |
| 400.0 | 3,500.0 | 13,000.0 | |
| 450.0 | 3,750.0 | 14,000.0 | |
| 500.0 | 4,000.0 | 15,000.0 | |
| 600.0 | 4,250.0 | 16,000.0 | |
| 700.0 | 4,500.0 | 17,000.0 | |
| 800.0 | 4,750.0 | | |
| 900.0 | | | |

