# Peer review of "Central-Pacific surface meteorology from the 2016 El Niño Rapid Response (ENRR) field campaign"

_Earth System Science Data, 2017_

## Author Comment (AC1) · 21 Dec 2017

We have just received a DOI for "El Niño Rapid Response (ENRR) Field Campaign: Surface Fluxes from the NOAA Ship Ronald H. Brown, 2016-02 to 2016-03 (NCEI Accession 0167875)", which is one of the three data sets discussed in this manuscript. The Copernicus staff has added it to the "Assets" page associated with this manuscript, thus making it easier for interested ESSDD readers to access. We will update the bibliographic and citation information in the manuscript later in the revision process.

---

## Referee Comment (RC1) · Anonymous Referee #1 · 31 Dec 2017

1. The dataset is new and useful for future investigation of the El Nino event as well as air-sea interactions. The authors did a good job on describing the instruments and the data processing. The detailed considerations of the data comparison and instrument calibrations are very helpful for future users. The methods for the data processing are standard. The references used are appropriate.

2. The link for obtaining the dataset works. I didn't download any data to check any downloading issues.

After the authors carefully discussed the surface pressure measurement and calibration, I don't see surface pressure listed in the final variable list in Table 5.

[Figure]

Unfortunate, no eddy-correlation turbulent fluxes are available. However using the sonic anemometers on the ship, the standard deviation of wind speed can be given in the final dataset, which can be very useful for representing turbulence intensity.

3. Data inconsistencies, if there are any, have to be discovered by users when detailed analyses are made.

4. The size of the hourly dataset should be manageable. Some users may want to get the dataset at higher sampling frequencies, such as every 5-min. Of course, they can contact the authors for such a possible dataset.

The article is well written. The good description and the easy access of the dataset will encourage scientists to use the dataset.

Here are some detailed comments and questions.

P.4, L.21. "both sites", I guess they are the island site and the ship site. Is this correct? P.6, L.25. So the radiosonde relative humidity is better? P.7, L.1. Was the RH/T sensor on the island aspirated? P.14, L.3. It is better to list variables calculated with the COARE flux algorithm in the final dataset here for those who are not familiar with the algorithm. Figure 4. It will be easy for readers to understand the dataset if the locations of the island and the buoys used in the dataset are marked in the figure. Figure 5. What does the "ground station" include here? Figure 6. Is "surface pressure" here the pressure at the island or on the deck? Soundings were launched on the desk of the ship only, is that correct? Figure 9. Is the "surface temperature" here the surface air temperature at the island?

---

## Short Comment (SC1) · 4 Jan 2018

I thank Anonymous Referee #1 for the review of our manuscript and suggestions for improvement. Before I post a point-by-point response on behalf of all authors, I would like to clarify something regarding the data involved.

This manuscript describes three data sets:

El Niño Rapid Response (ENRR) Field Campaign: Surface Meteorological and Ship Data from the NOAA Ship Ronald H. Brown, February-March 2016 (NCEI Accession 0161528) C. Cox, D. Wolfe, L. Hartten, and P. Johnston

https://doi.org/10.7289/V5SF2T80

El Niño Rapid Response (ENRR) Field Campaign: Surface Meteorological Data from Kiritimati Island, January-March 2016 (NCEI Accession 0161526) L. Hartten, P. Johnston, C. Cox, and D. Wolfe https://doi.org/10.7289/V51Z42H4

El Niño Rapid Response (ENRR) Field Campaign: Surface Fluxes from NOAA Ship Ronald H. Brown, 2016-02 to 2016-03 (NCEI Accession 0167875) C. Cox and L. Hartten https://doi.org/10.7289/V58050VP

The first two, containing surface meteorological data at high temporal resolution (1- or 2-minutes), were identified as "Assets" and shown under that when the manuscript was submitted. The third was not, because we had not yet received a DOI for it, although it was referenced in the manuscript by its NCEI accession number. In late December we received that DOI and the editorial staff at ESSD added the data set to the "Assets" tab.

---

## Referee Comment (RC2) · Anonymous Referee #2 · 25 Jan 2018

Review ESSD-2017-126

When possible use full doi prefix, e.g. https://doi.org/10.7289/V51Z42H4 in preference to doi: 10.7289/V51Z42H4?  For many users on certain browsers, the former offers one-click access while the later requires user intervention.

The NASA Ames (.na) format seems like a legacy format of the research aviation community, still used by, for example, NCAR.  The authors do a good job of explaining NASA Ames format in their Section 5 and a search in ESSD shows at least one other data set in NASA Ames format.  But international surface met data, as presented in several other ESSD data sets, often come in more generic formats, e.g. .tsv or .csv, or - maintaining full metadata - as NetCDF.  Provide a NetCDF version or mention a link to a translator?

NOAA download file includes useful help files.

Agree with the basic utility of these data on their own, and with the need to document logistical challenges both island and ship.  Good product, hope to see it widely used.  Recommend publication with some changes or additional explanations.

Specific comments

Page 4, line 5 "Global Class" research vessel.  NOAA or UNOLS designation?  Eliminate or explain for an international audience.

Page 6, line 4 "replaced suspicious data with flags."  In looking at the Kiritimati data file, I did not see flag values other than standard missing data designated by 9999.  Did the authors insert additional flag indicators?  If so, we should know their value and meaning?  If not, we should know that we can not distinguish them from other 9999. data?

Page 10, section 3.2.4 Winds: Here the authors ask us to accept a very large assumption, that winds and therefore intercomparisons and corrections from a different cruise of the same ship but much farther north (in a different wind regime) and one year earlier will also apply to the ENRR data. Without a working bow anemometer for the ENRR cruise the authors probably have little choice - and this reviewer well knows the unfortunately frequent disappointment of finding 'standard' ship met sensors not working as advertised - but some larger context would help.  In these strong ENSO or pre-strong ENSO conditions, the authors could give us a short (one paragraph?) summary of large scale wind, convection and SST conditions that would help our understanding and acceptance of wind, radiation and ocean temperature corrections?  Figure 4 provides a large scale picture of SST for the entire cruise of RV Brown but we could see similar fields of surface wind or pressure anomalies or of cloud top height / temperatures as an indication of convection?  If this time period represented an anomalous period for winds or convection then we have greater reason to worry about corrections based on 'mean' conditions or literature values?  In other words, other than for a very warm SST, do the authors consider this time period normal (for the location, season, ENSO index, etc.) or highly unusual?

Page 12, Section 4: Surface flux calculations.  Do we need a sentence that gives the explicit formulation of this version of the COARE algorithm?

After reading this section this review suspects some circular logic involved?  Use radiosonde data prior to launch to correct or validate some aspects of surface met data.  But (and presumably documented in the upper air data set) then use the surface met data to properly initialise the radiosonde data. Then, to calculate LWD contribution to fluxes, use the radiosonde data, at least for H2O, to estimate column profile of RH as it would influence LWD at the surface if measurements had included direct LWD.  Somehow this reviewer gets the uncomfortable feeling that upper air data corrected originally by surface data then became themselves an upper air input to a surface calculation?  Perhaps unavoidable but not ideal, deserves notice?  Also, rather than assuming vertical cloud distributions centered at 1 km (about optical depth I concede the authors assumptions), the radiosonde RH profiles probably indicate cloud layers, at least generally?  Or, would these assumptions and calculations prove insensitive to cloud layer height?

As I remember, NCAR and Vaisala originally published a correction to COARE radiosonde RH values, particularly to address erroneous surface dry layers. Now we use the radiosonde pre-launch surface RH values to calibrate surface RH sensors?

Page 12, line 9: Fairall et al 1996 does not represent the most recent version of the COARE bulk flux algorithm as implied in this sentence? Later the authors cite Edson 2013 for more recent versions?

The meat of this paper lies in the figures and tables. Tables excellent for the most part, very helpful. Figures likewise instructive and helpful, particularly the frequency distributions of various sensor differences. Although the text maintains a very good record of reporting uncertainties, the figures often fail to show uncertainty. Figure 13 represents a nice exception that proves the point? Showing uncertainty bands would make some figures unreadable? We need either explicit inclusion of uncertainties where appropriate or valid explanation of their absence? Perhaps particularly for figures 19 and 20?

Figure 8, RH does not go much higher than 95% during periods of heavy rain?

Figure 10, showing the rain events would also prove helpful here?

Figure 20, not clear why panel C (lat, lon) has relevance to upper two panels?

Table 6, vertical grid for RRTM: not sure the utility of this?

---

## Author Comment (AC2) · 27 Jan 2018

[12pt,a4paper]article [latin2]inputenc graphicx ulem amsmath  **Response to Referee #1 of essd-2017-126**

Referee comments are in *italics*; our responses follow each comment. A track-changes revised manuscript is uploaded as a supplement, and contains changes made in response to Anonymous Referee 1's comments as well as some edits we have made on our own.

*Referee #1:*

*General comment 2 (excerpt): After the authors carefully discussed the surface pressure measurement and calibration, I don't see surface pressure listed in the final variable list in Table 5.*

RESPONSE: One-minute values of the surface pressure from the NOAA Ship *Ronald H. Brown* are available in the ship's surface meteorology data set (Cox et al. 2017b, doi:10.7289/V5SF2T80) that is also documented in this manuscript. We did not include a one-hour version of the data in the surface flux data set because no changes had been made to the data beyond averaging. However, the reviewer's interest in such a time series is fairly easy to accommodate.

ACTION TAKEN: We have added hourly surface pressure, corrected to 3.8 m, to the surface flux data set and are in the process of submitting the revised file to NCEI. We have modified Table 5 and the associated text in section 4.3 accordingly.

*General comment 2 (excerpt): Unfortunate, no eddy-correlation turbulent fluxes are available. However using the sonic anemometers on the ship, the standard deviation of wind speed can be given in the final dataset, which can be very useful for representing turbulence intensity.*

RESPONSE: If wind data had been collected at 10 Hz or 20 Hz, sensible heat fluxes from eddy correlation would be possible, albeit with additional uncertainty compared to more conventional land-based observations because of the ship's motion (Fairall et al. 1996). In actuality, the ship's sonic anemometers operated at 1 Hz (c.f. Table 2), so any turbulence information that might otherwise be in the wind variance has been lost. Instead, estimations of both latent and sensible fluxes using bulk calculations were provided (Table 5). The algorithms used to make the calculations were developed for the tropical ocean and validated there in situ against direct measurements from eddy covariance (Fairall et al. 1996). The bulk fluxes should be considered significantly more robust than a proxy for intensity based on the standard deviation of the 1 Hz winds.

ACTION TAKEN: We have added clarifying statements to Section 4.

**Detailed comments:**

*. P.4, L.21.* *"both sites", I guess they are the island site and the ship site. Is this correct?*

RESPONSE: This is correct.

ACTION TAKEN: Reworded sentence and explicitly named the sites.

*. P.6, L.25.* *So the radiosonde relative humidity is better?*

RESPONSE: Yes, we believe that the radiosonde humidity values are better than the HMP45C humidity values because the humidity sensor in each radiosonde was calibrated during the ground check procedure. We are not claiming this as a general result, but as one specific to the measurements that we took during this campaign.

ACTION TAKEN: We have altered the beginning of the paragraph to explicitly declare our trust in the radiosonde humidities because of that calibration, and to tie our in-field awareness of HMP45C problems to that trust.

*. P.7, L.1.* *Was the RH/T sensor on the island aspirated?*

RESPONSE: The HMP45C was shielded from radiation (c.f. Table 1 and Figure 2) but not mechanically aspirated.

ACTION TAKEN: We have enhanced the shield's description in Table 1, and have added an explanatory footnote to the manuscript text that gave rise to the question.

*. P.14, L.3.* *It is better to list variables calculated with the COARE flux algorithm in the final dataset here for those who are not familiar with the algorithm.*

RESPONSE: We respectfully disagree. The surface flux data set referred to includes 43 variables; we list all of them in Table 5 (cited on P.14, L.6 of the submitted manuscript) and in that table also clearly identify the 30 that come from the COARE flux algorithm. To list them in the text would take many lines, would distract from the

purpose of the paragraph, and would, we believe, make it much harder for people to find the information.

ACTION TAKEN: We have added the following statement at the introduction to the COARE flux algorithm in section 4, before Table 5 is cited: "These calculations include an estimate of the full surface heat budget (see Section 4.3)".

*. Figure 4. It will be easy for readers to understand the dataset if the locations of the island and the buoys used in the dataset are marked in the figure.*

ACTION TAKEN: We have added a second panel to this figure, showing the locations of Kiritimati Island and of the buoys used in validating/correcting the surface data collected by the NOAA Ship *Ronald H. Brown*.

*. Figure 5. What does the "ground station" include here?*

RESPONSE: This refers to the radiosonde ground-check equipment. Much of our knowledge on how to collect data with these instruments has been transmitted orally and/or in a hands-on setting, with documentation primarily used for technical and procedural details. This has allowed our language to become imprecise. We appreciate the reviewer drawing this to our attention; it forced us to review the literature and sharpen the language in this manuscript as well as in the companion manuscript about our radiosonde data.

ACTION TAKEN: After reviewing the Vaisala literature's nomenclature, we have edited the annotation on Figure 5 to say "ground check set". We have also checked all in-text references to the radiosonde equipment and software, clarifying or correcting it to bring it into alignment with Vaisala usage, and adding some citations to the Vaisala literature in the process.

*. Figure 6. Is "surface pressure" here the pressure at the island or on the deck? Soundings were launched on the desk of the ship only, is that correct?*

RESPONSE: This figure uses data from Kiritimati Island. The ship-based soundings

were launched from the fantail.

ACTION TAKEN: We have improved the caption of Figure 6 in order to identify the source of the data.

*. Figure 9. Is the "surface temperature" here the surface air temperature at the island?*

RESPONSE: Yes, this figure uses data from Kiritimati Island.

ACTION TAKEN: We have improved the caption of Figure 9 in order to identify the source of the data.

[Figure]

**Supplement:**

[revised manuscript text omitted]
 we considered that to be our most trustworthy source of humidity information. In light of that, it became obvious very early in the project that there were problems with the station's original HMP45C humidity probe. These were not entirely solved by the replacement HMP45C. Figure 7a shows the HMP45C humidity readings with the co-located radiosonde data before the 14 February 2016 12 UTC launch. This was a very typical situation, with the HMP45C consistently reporting lower humidity than the radiosonde. Analysis of all co-located data collected after the HMP45C was replaced shows that the mean difference between the HMP45C and the co-located radiosonde humidity was 5.1% (Fig. 7b). We have therefore corrected the surface relative humidity for this bias by adding 5.1% to all reported relative humidity values that were considered good. The final values are shown in Fig. 8b.

During post-deployment evaluation of our data, we also compared the one-minute HMP45C temperature with one-minute averages of radiosonde temperatures collected while the radiosonde was co-located with the instrument tripod. All

comparisons are with the replacement sensor. The results (Fig. 9) show that the mean temperature difference between the radiosonde temperatures and those from the HMP45C is less than ±0.1 °C. This is better than expected, given the differences of shielding and aspiration[2]. We concluded that the temperature probe data, except for the initial deployment period when there was a bad probe, could be used as recorded in the field. Figure 8a shows the final time series.

**3.1.3 Winds**

Several hours after initial setup the base of the anemometer was aligned to face south using a Brunton hand-held compass. The user was standing within a few meters of the tripod and the anemometer was 3.88 m above ground level. Wind observations prior to this are of speed only. The alignment was repeated when the station was relocated. On 28 March 2016 the on-site observers used a hand-held compass and string to determine that the anemometer was aligned at 168° relative to magnetic north, within a ±3° range. Since the declination for our site is 9.03° east of north (National Centers for Environmental Information, 2016), the values recorded on site will show a 180° wind direction when the wind is 168°+9°=177°. We have therefore subtracted 3° from the recorded wind directions to correct for this bias. This correction was applied to the entire record, since the alignment method was the same during initial deployment and after the move. The measured wind speeds are the best estimate we have of the actual wind speeds, and we have not changed them. The final time series is shown in Fig. 10.

As mentioned above, neither the initial nor the final location of the surface instruments was ideal from a wind perspective. The bungalows were approximately 6 m high, and no place near our launch site qualified as "open terrain", i.e. with "the distance between the anemometer and any obstruction … at least 10 times … the height of the obstruction" (World Meteorological Organization, 2013, Part III Section 3.3.5). Therefore, even after the move there were still blockage issues affecting the measured winds. However, moving the station clearly increased the range of wind directions and speeds we observed. We have analysed the corrected winds over several time periods, calculating various statistics from the high-resolution time series and also evaluating the directions in both 5° and 10° bins[3]. Before the move, the mean wind speed was 2.23 m s$^{-1}$ with a mean direction of 22.5°; 71% of the winds came from 0°–50° (Fig. 11, left), with very small secondary clusters from the south (160°–180°, 5%) and southwest (230°–275°, 7%). Only 3.4% of the measured winds were from the 50°–160° sector, which is surprising considering Kiritimati's location in the trade wind region. After the move, the mean wind speed was 4.52 m s$^{-1}$ and the mean direction 61.9°; 85% of the winds came from 15°–100° (Fig. 11, right), with large clusters
* * *
[2] The HMP45C was inside a radiation shield, while the radiosonde was merely set in the shade of the solar panel. Neither the HMP45C nor the radiosonde were mechanically aspirated. The radiation shield is designed to allow aspiration by the ambient airflow, but we do not assume that the effects of air flow through the shield and air flow over the exposed radiosonde sensors was the same.

[revised manuscript text omitted]

Vaisala Oyj: User's Guide: Ground Check Set GC25, Vaisala Oyj, M210329EN-E, Helsinki, Finland, 43 pp., available from www.vaisala.com, 2006.

Vaisala Oyj: Vaisala Radiosonde RS92-SGP User's Guide, Vaisala Oyj, M210295EN-J, Helsinki, Finland, 54 pp., available from www.vaisala.com, 2015.

Woods Hole Oceanographic Institution (WHOI): ASIMET Module Specifications, http://frodo.whoi.edu/asimet/asimet_module_specs.html#bpr_mod (last access: 31 March 2017), 2010a.

Woods Hole Oceanographic Institution (WHOI): ASIMET Sensors, http://www.whoi.edu/instruments/viewInstrument.do?id=14686 (last access: 31 March 2017), 2010b.

World Meteorological Organization: Manual on the Global Observing System, 2010 ed., WMO (Series) no. 544, Secretariat of the World Meteorological Organization, Geneva, Switzerland, 2013.

Zipser, E. J.: The Line Islands Experiment, its place in tropical meteorology and the rise of the fourth school of thought, B. Am. Meteorol. Soc., 51, 1136-1146, 1970.

**Figures**

[Figure]

Figure 1: a) Map of Kiritimati showing the locations of Cassidy International Airport (PLCH) and CXENRR. Kiritimati map excerpted from U.S. Defense Mapping Agency (DMA) Stock No. 83BHA83130, courtesy of the University of Texas Libraries, The University of Texas at Austin. Map data from surveys by New Zealand (1938-1941) and the U.S. Navy (to 1962). Heights are in feet above mean high water springs; soundings are in fathoms (or fathoms and feet if less than 11 fathoms). b) Detail of the CXENRR site at the Captain Cook Hotel. The bungalow which served as our base of operations is in the center of the image, with markers to its north indicating the initial and final locations of the surface meteorological instruments.

[Figure]

[Figure]

**Figure 2: (a) The surface met station in its final position above the beach at Kiritimati. The anemometer sits atop the assembly, while the temperature/humidity sensor (left) and rain gauge (right) are attached to the crossbeam. Just below the crossbeam is the solar panel, and the box containing the Campbell datalogger, the power supply, and the barometer is mounted just below that. A plastic tube runs from the barometer out the tube on the bottom of the box, and can be seen mounted within a white disk just to the left of the solar panel. (b) The 26 March 2017 00 UTC radiosonde being put on the Campbell box in the shade of the solar panel prior to launch. The pressure tube egress and mount are clearly visible.**

[Figure]

**Figure 3: The original position of the surface instruments (left), and their location after being moved back towards the top of the beach on 5 Feb 2017 (right).**

[Figure]

a)

b)

**Figure 4:** (a) SSTs observed by the NOAA Ship *Ronald H. Brown* during ENRR, overlaid on SST anomalies. The latter are the departure from the ERA-Interim (Dee et al., 2011) 1979-2016 February through March mean. (b) The cruise track of the NOAA Ship Ronald H. Brown overlaid on a map showing the location of the TAO buoys visited during its cruise and also the location of Kiritimati Island. The buoys used in the post-processing of the ship's surface pressure, temperature, and humidity are highlighted.

[revised manuscript text omitted]

---

## Author Comment (AC3) · 23 Apr 2018

**Response to Referee #2 of essd-2017-126**

Referee comments are in *italics*; our responses follow each comment. A track-changes revised manuscript is uploaded as a supplement; it contains changes made in response to Anonymous Referee #2's comments on top of changes made in response to Anonymous Referee #1's comments and some edits we have made on our own. In the interests of reducing the size of the resulting file, we have "accepted" the deletion of several figures. However, the insertions of their replacements have not been

"accepted" and so are clearly marked.

**Referee #2 General comments:**

• **(excerpt):** *When possible use full doi prefix, e.g. https:// doi.org/ 10.7289/ V51Z42H4 in preference to doi:10.7289/V51Z42H4.*

RESPONSE: We checked with the ESSD editorial office and discovered that this is now the preferred method. We were unaware of that, and thank Referee #2 for drawing it to our attention.

ACTION TAKEN: Changed throughout.

• **(excerpt):** *The NASA Ames (.na) format seems like a legacy format of the research aviation community, still used by, for example, NCAR. The authors do a good job of explaining NASA Ames format in their Section 5 and a search in ESSD shows at least one other data set in NASA Ames format. But international surface met data, as presented in several other ESSD data sets, often come in more generic formats, e.g. .tsv or .csv, or - maintaining full metadata - as NetCDF. Provide a NetCDF version or mention a link to a translator?.*

RESPONSE: NASA Ames format is common enough to be one of the accepted formats at the Centre for Environmental Data Analysis (CEDA, formed in part from the British Atmospheric Data Centre (BADC)). We do not think that this format, which has space-delimited data below a header, differs significantly from tab-delimited (.tsv) or comma-delimited (.csv) files except for this: it has a metadata-rich header format which has stood the test of time, or at least the test of 20 years. Hartten has in the course of her career repeatedly developed her own ASCII data headers, inevitably finding them insufficient several years later, and is glad to take advantage of someone else's wellthought-out system.

We do not want to provide our data in multiple formats, as it greatly increases the overhead on our end and also increases the likelihood of version control issues. We have done some research into extant software that can convert NASA-Ames format to netCDF. The "NAPPy" software at https://github.com/cedadev/nappy was originally developed at BADC by Ag Stephens, who is now at CEDA. Two of our local data set experts have tried it with some of our ENRR files; they report that it does work, although the metadata contained in "special comment" and "normal comment" lines is concatenated in a fashion that is difficult to read by eye. The software is not python 3 compliant (A. Stephens, 2018, personal communication) and conversion between NASA-Ames format and netCDF is only available for unix or linux platforms. Another package that we found was built for that institution's very specific requirements and does not properly convert our data, nor does it save any of the metadata contained in our "special comment" and "normal comment" lines.

ACTION TAKEN: We have added text to Section 5 pointing interested users to the NAPPy software.

**Referee #2 Specific comments:**

• **Page 4, line 5.** *"Global Class" research vessel. NOAA or UNOLS designation? Eliminate or explain for an international audience.*

RESPONSE: We are not sure who designates vessels as being of a certain class, nor if the designations vary between countries. The NOAA Ship *Ronald H. Brown* is part of the (U.S.) Federal Oceanographic Fleet, and is described as Global Class by the (U.S.) Interagency Working Group on Facilities and Infrastructure (IWG-FI) and its predecessor body (see JSOST 2007, 2013, 2016). The U.S.A.'s University-National

Oceanographic Laboratory System (UNOLS) website, which lists it amongst the UN-OLS vessels (https://www.unols.org/ships-facilities/unols-vessels) as a "NOAA Global Class Vessel . . . scheduled in cooperation with UNOLS", seems to rely on external determinations of vessel class.

If the reviewer is looking for something that points to the specifications (e.g. size, capacity, speed, cruise duration) of a Global Class vessel, the cited JSOST documents make it clear that the definition of "Global Class" is evolving and no longer includes any numerical specifications of that nature.

Joint Subcommittee on Ocean Science and Technology (JSOST): Federal Oceanographic Fleet Status Report, Washington, D.C., 40, available at http://www.nopp.org/wp-content/uploads/2010/03/IWG-F-Fleet-Status-Report-Final.pdf, 2007.

Joint Subcommittee on Ocean Science and Technology (JSOST): Federal Oceanographic Fleet Status Report, Washington, D.C., 42, available at http://www.nopp.org/wp-content/uploads/2010/03/federal_oceanographic_fleet_status_report.pdf, 2013.

Joint Subcommittee on Ocean Science and Technology (JSOST): Federal Fleet Status Report: Current Capacity and Near-Term Priorities, Washington, D.C., 18, available at http://www.nopp.org/wp-content/uploads/2016/06/federal_fleet_status_report_final_03.2016.pdf, 2016.

ACTION TAKEN: Added a footnote giving the most recent applicable definition of Global Class from JSOST (2016) and added that document to the References.

• **Page 6, line 4.** *"replaced suspicious data with flags." In looking at the Kiritimati data file, I did not see flag values other than standard missing data designated by 9999. Did the authors insert additional flag indicators? If so, we should know their value and meaning? If not, we should know that we can not distinguish them from other 9999.*

*data?*

RESPONSE: This is a fair point.

ACTION TAKEN: Changed to "...have replaced suspicious data with the flags used to indicate missing data".

• **Page 10, section 3.2.4.** *Winds: Here the authors ask us to accept a very large assumption, that winds and therefore intercomparisons and corrections from a different cruise of the same ship but much farther north (in a different wind regime) and one year earlier will also apply to the ENRR data. Without a working bow anemometer for the ENRR cruise the authors probably have little choice ... but some larger context would help. In these strong ENSO or pre-strong ENSO conditions, the authors could give us a short (one paragraph?) summary of large scale wind, convection and SST conditions that would help our understanding and acceptance of wind, radiation and ocean temperature corrections? Figure 4 provides a large scale picture of SST for the entire cruise of RV Brown but we could see similar fields of surface wind or pressure anomalies or of cloud top height / temperatures as an indication of convection? If this time period represented an anomalous period for winds or convection then we have greater reason to worry about corrections based on 'mean' conditions or literature values? In other words, other than for a very warm SST, do the authors consider this time period normal (for the location, season, ENSO index, etc.) or highly unusual?*

RESPONSE: Our "intercomparisons and corrections from a different cruise of the same ship" concern only biases that are specifically associated with winds encountering the superstructure of the ship. This requires any sample of winds that encounter the ship (for example, the observations during CalWater), and requires no assumptions about the similarity in conditions between the two cruises. The only real assumption we are making is that the superstructure of the ship and the positions of the bridge anemometers are the same; to the best of our knowledge, this is true.

ACTION TAKEN: None.

• **Page 12, Section 4.** *Surface flux calculations. Do we need a sentence that gives the explicit formulation of this version of the COARE algorithm?*

ACTION TAKEN: We added a one-sentence comment about the algorithm in response to Anonymous Referee #1; we think that may help address your concern as well. We have also changed the citation to "Fairall et al., 1996, 2003; most recently updated in Edson et al. 2013".

• **Page 12, Section 4 (paragraph-long comments/questions broken into 4 segments for clarity).** *After reading this section this review suspects some circular logic involved? Use radiosonde data prior to launch to correct or validate some aspects of surface met data. But (and presumably documented in the upper air data set) then use the surface met data to properly initialise the radiosonde data.*

RESPONSE: Referee #2 is correct; surface observations are used to provide an estimate of the atmospheric state at the time and place of each radiosonde launch, and constitute the first point in the sounding. Those observations will affect the final sounding profiles of temperature, humidity, and winds in the lower part of the sounding. (The exact distances through which the effects occur are not documented in any literature we are aware of.) They will also have some affect throughout the depth of the pressure and/or height profiles if integration of the hypsometric equation is used to obtain either of those quantities. Ideally the surface observations are independent ones, but values from the radiosonde can also be used (Vaisala Oyj, 2010).

But we do not believe that we are using circular logic in our handling of the surface data

sets described in this manuscript.

Hartten et al. 2018 (formerly Hartten et al. 2017a) describes how we prepared radiosondes for launch. The first step is to use the Vaisala ground check set and the DigiCORA software to "recondition" the radiosonde's humidity sensors (to remove possible contaminants) and to then perform a "ground check", recalibrating the humidity sensors (against a built-in 0% relative humidity reference), the temperature sensor (against a built-in independent temperature reference), and the barometer (against a hand-entered independent pressure, if available).

With regards to the surface data sets, data from the radiosondes were only used as follows:

- to convince us to check the CXENRR barometer against a calibrated standard after the field campaign, which led us to adjust the surface pressure observations based on that recalibration and to use those values in the CXENRR data set described here;

- to detect a mean pressure bias in the WTEC observations that was a function of ship-relative wind direction, which led to no changes to the surface pressure observations presented in the data sets described here;

- to identify mean biases in the humidity measurements at both CXENRR and WTEC, which led us to remove those mean biases from the surface observations before creation of the data sets described here.

Putting all this information together with respect to humidity, which is the only variable in the surface data set that was changed based on comparisons with radiosonde

measurements: Each radiosonde's humidity sensors were calibrated against an independent transfer standard during the ground-check phase of pre-launch operations. At each site, we have identified a mean bias between more-or-less collocated humidity measurements made by those calibrated radiosonde sensors and our surface humidity instrument, and removed the mean bias from each site's surface humidity observations before including them in these data sets. In essence, during post-processing we calibrated our surface humidity instruments against a sequence of calibrated radiosonde humidity sensors.

Vaisala Oyj: DigiCORA® User's Guide, M210488EN-F, Vaisala Oyj, Helsinki, Finland, 113 pp., available from www.vaisala.com, 2010.

Updated reference for Hartten et al. (formerly 2017a):

Hartten, L. M., Cox, C. J., Johnston, P. E., Wolfe, D. E., Abbott, S., McColl, H. A., Quan, X.-W., and Winterkorn, M. G.: Ship- and island-based soundings from the 2016 El Niño Rapid Response (ENRR) field campaign, Earth Syst. Sci. Data Discuss., https://doi.org/10.5194/essd-2018-7, in review, 2018.

ACTION TAKEN: Added a summary sentence in Sections 3.1.2 and 3.2.3, and updated Hartten et al. (2017a) reference.

• **Page 12, Section 4 (con't).** *Then, to calculate LWD contribution to fluxes, use the radiosonde data, at least for H2O, to estimate column profile of RH as it would influence LWD at the surface if measurements had included direct LWD. Somehow this reviewer gets the uncomfortable feeling that upper air data corrected originally by surface data then became themselves an upper air input to a surface calculation? Perhaps unavoidable but not ideal, deserves notice?*

RESPONSE: Continuing from the previous response . . . We did use post-processed

WTEC radiosonde profiles of RH to estimate the column $H_2O$ profile that was one component of our surface LWD estimation process. Those radiosonde profiles were anchored by a surface humidity value, as is the usual practice for RS92 radiosondes; it is incorrect to say that the upper-air data were "corrected originally by surface data". As described above, that surface humidity value had been adjusted during postprocessing. The adjustment removed a long-term mean bias between our surface humidity sensor and a sequence of freshly calibrated humidity sensors. Such a comparison against a transfer standard is one classic way to calibrate an instrument; the fact that the freshly calibrated sensors were part of the radiosonde package is irrelevant.

Atmospheric profiles from RS92 radiosondes always include a surface measurement. Our text is very clear that our LWD estimation process used radiosonde profiles of both temperature and $H_2O$ profiles. The text also states that the "full COARE flux algorithm requires some atmosphere/ocean quantities that were not measured during the cruises", that "[u]nfortunately" LWD was among those, and that some variables the algorithm can compute were not produced because we did not have the quantities required to calculate them. We respectfully believe that this qualifies as clear notice that estimating LWD is not as good as measuring LWD would have been, but that it was necessary so that we could produce an estimate of the net surface heat flux.

ACTION TAKEN: None.

• **Page 12, Section 4 (con't).** *Also, rather than assuming vertical cloud distributions centered at 1 km (about optical depth I concede the authors assumptions), the radiosonde RH profiles probably indicate cloud layers, at least generally? Or, would these assumptions and calculations prove insensitive to cloud layer height?*

RESPONSE: We visually inspected each sounding and subjectively estimated a cloud

base height, as you suggested. We found it quite hard to do. (Estimating cloud top would have been easier because of the cloud-top inversion, but cloud base was more important for the calculations performed here.) Our estimates looked plausible, with higher cloud bases to the north of the warm pool and lower ones (close to the marine boundary layer) near the equator, and the average of the estimated heights was 0.82 km, which is quite close to the nominal height of 1 km used in the manuscript. However, we are not particularly comfortable with these cloud-base height estimates and are not prepared to defend them, so we would not want to use them individually, nor in average or median form.

We have therefore tested the LWD sensitivity to cloud layer height by re-calculating the fluxes for various cloud heights, assuming bases from 0.25 km by 0.25 km to 5 km (193 sondes at 20 heights = 3860 calculations). As stated in the manuscript, the mean cloud radiative effect LWCRE for a cloud base at 1 km, averaged over all 193 radiosondes, is 31.1 W m$^{-2}$. Because the cloud temperature gets colder as height increases, the cloud forcing (and thus LWD) decreases with increasing cloud height at a rate of about –2.7 W m$^{-2}$ per extra 1 km of height. While small, relative to the estimated cloud forcing, it is not negligible (about 9% of cloud forcing per km).

ACTION TAKEN: We have added text clarifying why we assumed the cloud bases were at 1 km, and have also briefly noted the sensitivity of our results to that assumption.

• **Page 12, Section 4 (con't).** *As I remember, NCAR and Vaisala originally published a correction to COARE radiosonde RH values, particularly to address erroneous surface dry layers. Now we use the radiosonde pre- launch surface RH values to calibrate surface RH sensors?*

RESPONSE: Yes, we do. Referee #2 is correct that NCAR and Vaisala developed a set of humidity corrections for the RS80 radiosondes used in COARE (Wang et al.

2002) that addressed low-level dry biases among other things. For ENRR we used Vaisala RS92 radiosondes, which have different humidity sensors (Wang et al. 2013), together with DigiCORA sounding software v3.64.1 (Vaisala Oyj, 2010), which includes a humidity correction for the daytime solar radiation dry bias identified in the RS92 by Vömel et al. (2007). Ciesielski et al. (2014) determined that tropical soundings collected over the Indian Ocean and west Pacific warm pool during the Dynamics of the Madden-Julian Oscillation (DYNAMO) field campaign at "sites using VRS92 sondes with D3.64 software need[ed] no further humidity corrections".

Note also that Wang et al. (2002) say the following in the final paragraph of their article: "Comparisons of prelaunch radiosonde data with the surface data from independent surface sensors can always be used to evaluate the accuracy of radiosonde data (and/or surface data), and may provide some guidance on how to correct the data."

Ciesielski, P. E., Yu, H., Johnson, R. H., Yoneyama, K., Katsumata, M., Long, C. N., Wang, J., Loehrer, S. M., Young, K., Williams, S. F., Brown, W., Braun, J., and Van Hove, T.: Quality-Controlled Upper-Air Sounding Dataset for DYNAMO/CINDY/AMIE: Development and Corrections, Journal of Atmospheric and Oceanic Technology, 31, 741-764, https://doi.org/10.1175/jtech-d-13-00165.1, 2014.

Vömel, H., Selkirk, H., Miloshevich, L., Valverde-Canossa, J., Valdés, J., Kyrö, E., Kivi, R., Stolz, W., Peng, G., and Diaz, J. A.: Radiation Dry Bias of the Vaisala RS92 Humidity Sensor, Journal of Atmospheric and Oceanic Technology, 24, 953-963, https://doi.org/10.1175/jtech2019.1, 2007.

Wang, J., Cole, H. L., Carlson, D. J., Miller, E. R., Beierle, K., Paukkunen, A., and Laine, T. K.: Corrections of Humidity Measurement Errors from the Vaisala RS80 Radiosonde—Application to TOGA COARE Data, Journal of Atmospheric and Oceanic Technology, 19, 981-1002, https://doi.org/10.1175/1520-0426(2002)019<0981:Cohmef>2.0.Co;2, 2002.

Wang, J., Zhang, L., Dai, A., Immler, F., Sommer, M., and Vömel, H.: Radiation Dry Bias Correction of

Vaisala RS92 Humidity Data and Its Impacts on Historical Radiosonde Data, Journal of Atmospheric and Oceanic Technology, 30, 197-214, https://doi.org/10.1175/jtech-d-12-00113.1, 2013.

ACTION TAKEN: We have added the quote from Wang et al. (2002) to section 3.1, where the issue first arises. Citations to Vaisala documentation of the RS92 radiosondes and the ground check were added in response to comments from Referee #1.

• **Page 12, line 9.** *Fairall et al 1996 does not represent the most recent version of the COARE bulk flux algorithm as implied in this sentence? Later the authors cite Edson 2013 for more recent versions?*

ACTION TAKEN: Changed citation to read "Fairall et al., 1996, 2003; most recently updated in Edson et al. 2013".

• **Various.** *Although the text maintains a very good record of reporting uncertainties, the figures often fail to show uncertainty. Figure 13 represents a nice exception that proves the point? Showing uncertainty bands would make some figures unreadable? We need either explicit inclusion of uncertainties where appropriate or valid explanation of their absence? Perhaps particularly for figures 19 and 20?*

RESPONSE: Throughout the manuscript we have plotted data in ways that we felt best helped us to tell our story; sometimes plots lent themselves to also visualizing uncertainty, and sometimes they did not (for instance, when uncertainty bands would be indistinguishable on the plot scale, or when they would render portions of the plot unreadable). We agree that explicitly including uncertainties for our estimated LWD and our computed $Q_0$ is important.

ACTION TAKEN: Added Table 7 and text to Sections 4.2 and 4.3 to summarize estimated uncertainties in the individual heat budget terms, as well as the combined

[Figure]

surface heat budget uncertainty, which we estimate to be 20.8 W m$^{-2}$. The largest sources of uncertainty are from the assumed albedo (0.055) and the calculation of LWD.

- **Figure 8.** *RH does not go much higher than 95% during periods of heavy rain?*

RESPONSE: That is correct, and commonplace in the tropics (C. Fairall 2018, personal communication; M. A. LeMone 2018, personal communication) unless there is fog (P. E. Ciesielski 2018, personal communication). Figure 1 (below) shows the relative humidity and accumulated precipitation measured aboard the R/V Roger Revelle during three cruise legs of the DYNAMO field campaign in the equatorial Indian Ocean (Yoneyama et al. 2013). Leg 1 is omitted because only 35 mm of rain was recorded during the three-week long cruise. During Legs 2–4, even when rainfall was heavy surface humidities were usually below 95%, and any excursions above 95% were short-lived.

Yoneyama, K., Zhang, C., and Long, C. N.: Tracking pulses of the Madden–Julian Oscillation, B. Am. Meteorol. Soc., 94, 1871-1891, https://doi.org/10.1175/bams-d-12-00157.1, 2013.

ACTION TAKEN: None.

- **Figure 10.** *showing the rain events would also prove helpful here*

ACTION TAKEN: Shading indicating the heavy rain events has been added.

- **Figure 20.** *not clear why panel C (lat, lon) has relevance to upper two panels*

RESPONSE: It is provided as a reference for those who might want to get a quick look at how $Q_0$, $T$, and/or SST vary with ship position.

ACTION TAKEN: None.

- **Table 6.** *vertical grid for RRTM: not sure the utility of this*

RESPONSE: It is provided as a reference for those who might want to know the atmospheric structure our estimation method could possibly contain.

ACTION TAKEN: None

**Supplement:**

[revised manuscript text omitted]

* * *
[2] "Global Class ships are the largest and most capable with the ability to work worldwide with large scientific parties and the longest endurance" (Joint Subcommittee on Ocean Science and Technology (JSOST), 2016).

correct them were different for each site and are described below. Only those measured quantities that required detailed post-deployment analysis are discussed in subsections; all other information is given in the main site sections.

**3.1 Kiritimati Island**

Issues in the original Kiritimati Island surface data fall into five categories: instrument calibration issues; instrument failures;
5   non-representativeness of the data caused by poor siting; changes in the data acquisition methods and archival timing; and gaps in the data. Some of these issues were consequences of the "Rapid Response" aspect of the project. Because there is only one international flight to Kiritimati Island each week and the campaign was pulled together in only a few months, there was no time for a site survey prior to the start of fieldwork. This limited the choices of available sites and led to some of the shipped equipment being our "best guess" at what would be needed. It also meant that we often had to make do with what
10   was available on the island, or wait a week or more until something could be shipped.

The "Rapid Response" also affected the exact instruments deployed, which had to be those available on short notice from PSD's instrument pool. Questions about the calibration of some of them arose soon after the campaign started, which led us to develop a practice of placing the initialized Vaisala RS92 radiosonde atop the instrument box for an extended period before
15   launch. Radiosonde initialization, which was done inside the air-conditioned bungalow, includes a  ground-check procedure that uses high-precision and high-accuracy temperature and humidity sensors to calibrate each radiosonde (Vaisala Oyj, 2006, 2015). Setting the radiosonde in the shade of the surface station's solar panel (Fig. 2b) allowed the radiosonde to equilibrate to the outside environment; it also provided as much as several minutes of co-located pressure, temperature, and humidity data from the radiosonde and the surface instruments. Details of how we used these coincident measurements to
20   refine the surface observations are discussed in the relevant subsections below, and are in accord with Wang et al.'s (2002) suggestion that "comparisons of prelaunch radiosonde data with the surface data from independent surface sensors can always be used to evaluate the accuracy of radiosonde data (and/or surface data), and may provide some guidance on how to correct the data".

25   There were two instrument failures during the experiment. During the second week of operations it became clear that the temperature/humidity probe had failed. Fortunately, we had brought a second unit with us. We swapped the sensors on 6 February and finalized the replacement sensor's settings on 7 February 2016 at 03:41 UTC; it operated well for the rest of the campaign. On 15 March 2016, at 11:12 UTC, the solar panel providing power for the data logger failed. It took about 18 hours to diagnose the problem, find a replacement power supply and install it, and restart the surface station.

The surface observations were critical for the initialization of the radiosondes, so the surface station had to be located near the radiosonde release location, i.e. close to the bungalow housing the radiosonde data system. The surface station was originally

set up on a grassy area northeast of the concrete pad. This location looked good at first, but after a few days of operations the staff realized that the anemometer sometimes spun in circles, indicating swirling winds; that the range of recorded (2-minute) wind directions was quite narrow; and that recorded winds were sometimes inconsistent with the values reported by the radiosondes after they had risen above the bungalows and trees. The surface station was moved to a better location, at the edge of the beach and further from all the bungalows, on 5 February 2016 (21-22 UTC).

On 11 February 2016, at 04:05:43 UTC, four changes were made to the data logger program. The execution interval, which determines how often each instrument is read, was changed from 2 seconds to 1 second. The internal offset for the barometer was changed from 599.5 to 399.5, so that pressure would be reported with a resolution of 0.01 hPa instead of 0.1 hPa. The averaging time was changed from 2 minutes to the more standard 1 minute. All these changes were made to give better data to help with radiosonde surface initialization and with data evaluations.

The failure of the T/RH sensor, the station move, the change to the data logger software, and the failure of the data logger power supply all caused gaps in the data (Table 3). We have not attempted to interpolate across any of the data gaps, nor have we tried to interpolate the two-minute data to one-minute values. Instead we have carefully examined data recorded before and after instrument moves, failures, and replacements, and have replaced suspicious data with the flags used to indicate flagsmissing data.

There is one final general note about the surface meteorology data. The radiosonde system maintains time very accurately, but it does so by getting the time from GPS satellites. The DigiCORA sounding software (Vaisala Oyj, 2010; version 3.64.1)Vaisala Digicora software sets the computer clock to GPS time, which was 17 s ahead of UTC time during ENRR, and the Campbell software sets the data logger clock to the computer's time. For the final Kiritimati surface data set, all times have been shifted by 17 s, so that all observations are reported in UTC.

**3.1.1 Surface Pressure**

The radiosonde ground ground-check procedure requires an external pressure reading to provide a calibration check and correction to the radiosonde pressure sensor. This was provided by the Vaisala PTB101B pressure sensor mounted at 3.6 m ASL on the instrument tripod. The barometer was installed with a Vaisala SPH10 static pressure head to minimize wind induced errors. During the first month of the experiment, the staff noted pressure differences between the PTB101B and both the radiosondes (Fig. 6) and other barometers. The PTB101B was checked against a high-quality calibrated standard after returning to Colorado. The PTB101B read 0.64 hPa too high, so in the final data set all the surface pressure data was decreased by 0.64 hPa to account for this calibration offset.

**3.1.2 Relative Humidity and Temperature**

The radiosonde  ground-check procedure serves to calibrate each radiosonde's humidity sensor, so we considered that to be our most trustworthy source of humidity information. In light of that, it became obvious very early in the project that there were problems with the station's original HMP45C humidity probe. These were not entirely solved by the replacement HMP45C. Figure 7a shows the HMP45C humidity readings with the co-located radiosonde data before the 14 February 2016 12 UTC launch. This was a very typical situation, with the HMP45C consistently reporting lower humidity than the radiosonde. Analysis of all co-located data collected after the HMP45C was replaced shows that the mean difference between the HMP45C and the co-located radiosonde humidity was 5.1% (Fig. 7b). We have therefore corrected the surface relative humidity for this bias by adding 5.1% to all reported relative humidity values that were considered good. (Put another way, we calibrated our replacement HMP45C against a sequence of calibrated radiosonde humidity sensors.) The final values are shown in Fig. 8b.

During post-deployment evaluation of our data, we also compared the one-minute HMP45C temperature with one-minute averages of radiosonde temperatures collected while the radiosonde was co-located with the instrument tripod. All comparisons are with the replacement sensor. The results (Fig. 9) show that the mean temperature difference between the radiosonde temperatures and those from the HMP45C is less than ±0.1 °C. This is better than expected, given the differences of shielding and aspiration[3]. We concluded that the temperature probe data, except for the initial deployment period when there was a bad probe, could be used as recorded in the field. Figure 8a shows the final time series.

**3.1.3 Winds**

Several hours after initial setup the base of the anemometer was aligned to face south using a Brunton hand-held compass. The user was standing within a few meters of the tripod and the anemometer was 3.88 m above ground level. Wind observations prior to this are of speed only. The alignment was repeated when the station was relocated. On 28 March 2016 the on-site observers used a hand-held compass and string to determine that the anemometer was aligned at 168° relative to magnetic north, within a ±3° range. Since the declination for our site is 9.03° east of north (National Centers for Environmental Information, 2016), the values recorded on site will show a 180° wind direction when the wind is 168°+9°=177°. We have therefore subtracted 3° from the recorded wind directions to correct for this bias. This correction was applied to the entire record, since the alignment method was the same during initial deployment and after the
* * *
[3] The HMP45C was inside a radiation shield, while the radiosonde was merely set in the shade of the solar panel. Neither the HMP45C nor the radiosonde were mechanically aspirated. The radiation shield is designed to allow aspiration by the ambient airflow, but we do not assume that the effects of air flow through the shield and air flow over the exposed radiosonde sensors was the same.

[revised manuscript text omitted]

25   **4 Surface Fluxes at the NOAA Ship *Ronald H. Brown***

Based on the ENRR surface observations described in the previous section, we calculated bulk air-sea fluxes (1-hour averages) using a recent version of the Tropical Ocean and Global Atmosphere (TOGA) Coupled Ocean–Atmosphere Response Experiment (COARE) flux algorithm (Fairall et al., 1996, 2003; most recently updated in Edson et al. 2013). These calculations include an estimate of the full surface heat budget (see Section 4.3). A complete list of the input data and the

30   variables calculated from it by the COARE flux algorithm are in Table 5. The full COARE algorithm requires some atmosphere/ocean quantities that were not measured during the cruises. With the exception of longwave radiation (LWD) we

 We have attempted to estimate radiative fluxes that were not measured
directly using other variables that were, as described below, but some other variables such as rain rate and direct measurements
of turbulent fluxes from eddy covariance methodology are not provided because the ancillary measurements are not available.

5 However, bulk calculations of both latent and sensible heat fluxes are made using COARE and are provided (Table 5). The
bulk calculations from COARE have been previously validated in the tropical ocean by comparison to direct measurements
from eddy covariance (Fairall et al., 1996). Longwave downwelling radiation (LWD) is needed for the bulk flux calculations
within the COARE algorithm if a cool-skin correction is applied to SST; it is required if a full surface heat budget is to be
calculated from the COARE results. Unfortunately, LWD was not measured during this cruise and so we estimated it using

10 other means, as described in 4.2.

**4.1 Treatment of Measured Properties**

[revised manuscript text omitted]

30  2.7 W m$^{-2}$ per extra 1 km of cloud base height (i.e. the mean LWCRE would be 4.3% smaller (larger) if the assumed cloud base were at 1.5 km (0.5 km)). We performed a second sensitivity study whereinin which we recalculated LWCRE after assigning the cloud bases to reasonable heights based on visual analysis of the radiosonde profiles. Estimating cloud bases in that manner was quite challenging, but while we do not have enough confidence in the estimates to use them in our flux

calculation we think they can put bounds on the possible errors in our 1 km assumption.  Comparing those LWCRE values  with the ones based on a nominal 1 km cloud height  revealed maximum differences of  ±6 W m$^{-2}$, nearly 20% of LWCRE on average, which we take to be a reasonable estimate of the error in LWCRE.

**4.3 Bulk Flux Calculations and Final Data Set**

5    We used the pressure, air temperature, wind, bulk ocean temperature, solar radiation, and longwave radiation time series described above as input to the COARE flux algorithm (version coare35vnWarm; Edson et al., 2013). In keeping with Fairall et al. (1996), we assumed a surface emissivity of 0.97 and a surface albedo of 0.055. The resulting time series, together with a subset of the surface time series used as input, constitute the released bulk flux data set (Table 5). we assumed a surface emissivity of 0.97 and a surface albedo of 0.045.

10

15

20     a subset of the surface time series used as input constitute the released bulk flux data  Figure 20a shows the net surface heat flux $Q_0$ calculated from some of the measurements and calculated quantities as

$$Q_0 = Q_{sw} + Q_{lw} - Q_{lat} - Q_{sen} , \tag{2}$$

25   where $Q_{sw}$ is the net (absorbed) solar irradiance, $Q_{lw}$ is the net longwave flux, $Q_{lat}$ is the bulk latent heat flux out of the ocean and $Q_{sen}$ is the bulk sensible heat flux out of the ocean. For reference, both the observed air temperature and ocean temperature are plotted in Fig. 20b. The large amplitude of the diurnal cycle (500-1000 Wm$^{-2}$) indicates that uncertainty associated with the estimate of LWD is quite small relative to the magnitude of the other terms except for the times of the day when the net surface heat budget is switching from net warming to net cooling, and *vice versa* (near sunrise and sunset).

Since the true values of surface emissivity and surface albedo are not precisely known, nor would they be constant, our assumed values result in uncertainty in the calculations of upwelling longwave and shortwave radiation (LWU and SWU), respectively.

The uncertainty in the calculation of LWU associated with the estimation of skin temperature, as well as a range of emissivity values from the literature, are also shown in Table 7. Sensitivity studies using the ocean albedo parameterization of Jin et al. (2011) suggest 0.055 is a poor approximation for clear skies when albedo is dependent on solar zenith angle but is a close approximation for the diffuse regime (overcast skies), so our usage is consistent with the high cloud fraction during the cruise.

5    Unfortunately, we do not have a good estimate of the diffuse fraction to use for a calculation of albedo, but by comparing an overcast and clear-sky albedo parameterization using the Jin et al. routines, we have added the contribution of cloudiness to the uncertainty in SWU (Table 7). Since albedo is also dependent on wind speed (due to its effect on surface roughness), we also estimated an uncertainty in diffuse-regime albedo by comparing estimates including wind speeds with errors of ±1 m s$^{-1}$ to albedo = 0.055, reporting the largest error (RMSE) of the three-way comparison in calculated SWU (Table 7). Estimates of

10   uncertainty in the other variables used to calculate $Q_0$ are also shown in Table 7.;. We estimate the combined uncertainty in $Q_0$ we estimate to be 20.7 W m$^{-2}$.

**5 Data Availability and Use**

The data sets described here are archived at NOAA's National Center for Environmental Information (NCEI), with DOIs as follows: surface meteorology from Kiritimati (https://doi.org/doi:10.7289/V51Z42H4), surface meteorology and SST from the

15   NOAA Ship *Ronald H. Brown* (https://doi.org/doi:10.7289/V5SF2T80), and surface fluxes from the NOAA Ship *Ronald H. Brown* (https://doi.org/10.7289/V58050VPNCEI Accession Number 0167875, available at http://accession.nodc.noaa.gov/0167875). They are also available, together with sample code to read the data, from NOAA/ESRL/PSD at https://www.esrl.noaa.gov/psd/enso/rapid_response/data_pub/. Users of these data must cite the appropriate DOI and reference the data as indicated below:

20   Cox, C., and Hartten, L.: El Niño Rapid Response (ENRR) Field Campaign: Surface Fluxes from the NOAA Ship Ronald H. Brown, 2016-02 to 2016-03 (NCEI Accession 0167875), Version 1.1, NOAA /National Centers for Environmental Information, https://doi.org/10.7289/V58050VPdoi pending, 2017.

Cox, C., Wolfe, D., Hartten, L., and Johnston, P.: El Niño Rapid Response (ENRR) Field Campaign: Surface Meteorological

25   and Ship Data from the NOAA Ship Ronald H. Brown, February-March 2016 (NCEI Accession 0167875), Version 1.1, NOAA /National Centers for Environmental Information, https://doi.org/doi:10.7289/V5SF2T80, 2017.

Hartten, L., Johnston, P., Cox, C., and Wolfe, D.: El Niño Rapid Response (ENRR) Field Campaign: Surface Meteorological Data from Kiritimati Island, January-March 2016 (NCEI Accession 0161526), Version 1.1, NOAA /National Centers for

30   Environmental Information, https://doi.org/doi:10.7289/V51Z42H4, 2017.

The data from Kiritimati start on 25 January 2016 03:07:43 UTC and end on 28 March 2016 19:14:43 UTC, with four gaps in one or more parameters lasting an hour or more (Table 3). The ship data start on 16 February 2016 18:32 UTC and end on 16 March 2016 from 18:50–18:57 UTC, depending on the parameter.

Each data set is in the form of a single file written in the NASA Ames Format for Data Exchange (hereafter "NASA-Ames Format"); see Gaines and Hipskind (1998) as well as the Centre for Environmental Data Analysis (CEDA)British Atmospheric Data Centre (BADC) explanatory material (Centre for Environmental Data AnalysisBritish Atmospheric Data Centre, 2002).

5 NASA-Ames Format's plain-text (ascii) nature makes it portable to any machine and easily accessible to users with limited computer resources; the rich metadata in the mandated mandatory and optional header sections make it self-describing. NASA-Ames Format requires that the total number of header lines be the first number in the first line of the file, and that the data following the header lines be space-delimited. Our files use the last header line to provide rudimentary column headers for the data that follow, which makes importing the data into spreadsheets and starting to work with them fairly straightforward.

10 Time is the independent variable in all the files, and is presented as days since 1900-01-01 00:00:00 +00:00, i.e. since 1 January 1900 00:00:00 UTC. Users working with the data from Kiritimati Island, especially those interacting with residents and local records, should keep in mind that Kiritimati is in the Line Islands Time (LINT) time zone, UTC+14. Users who prefer to work with data in netCDF format may build their own routines to do so, or may use software such as NASA Ames Processing in PYthon (NAPPy, available at https://github.com/cedadev/nappy). We would caution users to carefully cross-check the output

15 of any conversion software, to make sure that it correctly puts into the netCDF file(s) all the original data as well as the metadata contained in the "special comment" and "normal comment" header sections.

[revised manuscript text omitted]

Centre for Environmental Data Analysis (CEDA): NASA Ames Format for Data Exchange, https:// http://cedadocs.ceda.ac.uk/73/4/index.html (last access:  18  January 2018), 2002.

20    Climate Prediction Center (CPC)   and   International Research Institute for Climate and Society (IRI):  El  Niño/Southern Oscillation                          (ENSO)                          Diagnostic                          Discussion, http://www.cpc.ncep.noaa.gov/products/analysis_monitoring/enso_disc_jun2015/ensodisc.html (last access: 2 March 2017), 2015.

25    Conley, S. A., Faloona, I., Miller, G. H., Lenschow, D. H., Blomquist, B., and Bandy, A.: Closing the dimethyl sulfide budget in the tropical marine boundary layer during the Pacific Atmospheric Sulfur Experiment, Atmos. Chem. Phys., 9, 8745-8756, https://doi.org/10.5194/acp-9-8745-2009, 2009.

Cox, C. and Hartten, L.: El Niño Rapid Response (ENRR) Field Campaign: Surface Fluxes from  NOAA Ship Ronald H.
30    Brown, 2016-02 to 2016-03 (NCEI Accession 0167875), Version 1.1, NOAA /National Centers for Environmental Information, https://doi.org/10.7289/V58050VP, 2017.

Cox, C. J., Walden, V. P., Rowe, P. M., and Shupe, M. D.: Humidity trends imply increased sensitivity to clouds in a warming Arctic, Nat. Commun., 6, 10117, https://doi.org/10.1038/ncomms10117, 2015.

Cox, C., Wolfe, D., Hartten, L., and Johnston, P.: El Niño Rapid Response (ENRR) Field Campaign: Radiosonde Data (Level 2) from the NOAA Ship Ronald H. Brown, February-March 2016 (NCEI Accession 0161527), Version 1.1, NOAA /National Centers for Environmental Information, https://doi.org/10.7289/V5X63K15, 2017a.

Cox, C., D. Wolfe, L. Hartten, and P. Johnston: El Niño Rapid Response (ENRR) Field Campaign: Surface Meteorological and Ship Data from the NOAA Ship Ronald H. Brown, February-March 2016 (NCEI Accession 0161528), Version 1.1, NOAA /National Centers for Environmental Information, https://doi.org/10.7289/V5SF2T80, 2017b.

5  Dee, D. P., Uppala, S. M., Simmons, A. J., Berrisford, P., Poli, P., Kobayashi, S., Andrae, U., Balmaseda, M. A., Balsamo, G., Bauer, P., Bechtold, P., Beljaars, A. C. M., van de Berg, L., Bidlot, J., Bormann, N., Delsol, C., Dragani, R., Fuentes, M., Geer, A. J., Haimberger, L., Healy, S. B., Hersbach, H., Hólm, E. V., Isaksen, L., Kållberg, P., Köhler, M., Matricardi, M., McNally, A. P., Monge-Sanz, B. M., Morcrette, J.-J., Park, B.-K., Peubey, C., de Rosnay, P., Tavolato, C., Thépaut, J.-N., and Vitart, F.: The ERA-Interim reanalysis: configuration and performance of the data assimilation system, Q. J. R. Meteorol. Soc.,
10  137, 553-597, https://doi.org/10.1002/qj.828, 2011.

Dole, R. M., Spackman, J. R., Newman, M., Compo, G. P., Smith, C. A., Hartten, L. M., Barsugli, J. J., Webb, R. S., Hoerling, M. P., Cifelli, R., Wolter, K., Barnet, C. D., Gehne, M., Gelaro, R., Kiladis, G. N., Abbott, S., Albers, J., Brown, J. M., Cox, C. J., Darby, L., de Boer, G., DeLuisi, B., Dias, J., Dunion, J., Eischeid, J., Fairall, C., Gambacorta, A., Gorton, B., Hoell, A.,
15  Intrieri, J., Jackson, D., Johnston, P. E., Konopleva-Akish, E. A., Lataitis, R., Mahoney, K. M., McCaffrey, K., McColl, H. A., Mueller, J. J., Murray, D., Neiman, P. J., Otto, W., Persson, O., Quan, X.-W., Rangwala, I., Ray, A. J., Reynolds, D., Dellaripa, E. R., Rosenlof, K., Sakaeda, N., Sardeshmukh, P. D., Slivinski, L. C., Solomon, A., Smith, L., Swales, D., Tulich, S., White, A., Wick, G., Winterkorn, M. G., Wolfe, D. E., and Zamora, R.: Advancing Science and Services during the 2015-16 El Niño:  The El Niño Rapid Response Field Campaign, B. Am. Meteorol. Soc., in press, https://doi.org/10.1175/BAMS-
20  D-16-0219.1, 2018.

Edson, J. B., Zappa, C. J., Ware, J. A., McGillis, W. R., and Hare, J. E.: Scalar flux profile relationships over the open ocean, J. Geophys. Res.-Oceans, 109, https://doi.org/10.1029/2003JC001960, 2004.

25  Edson, J. B., Jampana, V., Weller, R. A., Bigorre, S. P., Plueddemann, A. J., Fairall, C. W., Miller, S. D., Mahrt, L., Vickers, D., and Hersbach, H.: On the Exchange of Momentum over the Open Ocean, J. Phys. Oceanogr., 43, 1589-1610, https://doi.org/10.1175/jpo-d-12-0173.1, 2013.

Evans, M. N., Fairbanks, R. G., and Rubenstone, J. L.: The thermal oceanographic signal of El Niño reconstructed from a
30  Kiritimati Island coral, J. Geophys. Res.-Oceans, 104, 13409-13421, https://doi.org/10.1029/1999JC900001, 1999.

Fairall, C. W., Bradley, E. F., Rogers, D. P., Edson, J. B., and Young, G. S.: Bulk parameterization of air-sea fluxes for Tropical Ocean-Global Atmosphere Coupled-Ocean Atmosphere Response Experiment, J. Geophys. Res.-Oceans, 101, 3747-3764, https://doi.org/10.1029/95JC03205, 1996.
35
Fairall, C. W., Bradley, E. F., Hare, J. E., Grachev, A. A., and Edson, J. B.: Bulk parameterization of air-sea fluxes:  Updates and verification for the COARE algorithm, J. Climate, 16, 571-591, https://doi.org/10.1175/1520-0442(2003)016%3C0571:BPOASF%3E2.0.CO;2, 2003.

40  Fisher, N. I.: Statistical Analysis of Circular Data, Cambridge University Press, Cambridge, 277 pp., 1995.

Fisher, N. I.: Further erratum for Statistical Analysis of Circular Data, available at: http://www.valuemetrics.com.au/Downloads/Directional/directional.zip, last access: 28 August 2017, 2017.

45  Gage, K. S., Balsley, B. B., Ecklund, W. L., Carter, D. A., and McAfee, J. R.: Wind profiler-related research in the tropical Pacific, J. Geophys. Res.-Oceans, 96, 3209-3220, https://doi.org/10.1029/90JD01829, 1991.

Gaines, S. E., and Hipskind, R. S.: Format Specification for Data Exchange, Version 1.3, NASA Ames Research Center, 30, available at: http://cloud1.arc.nasa.gov/solve/archiv/archive.tutorial.html, 1998.

Hartten, L. M., Cox, C. J., Johnston, P. E., Wolfe, D. E., Abbott, S., McColl, H. A., Quan, X.-W., and Winterkorn, M. G.: Ship- and island-based soundings from the 2016 El Niño Rapid Response (ENRR) field campaign, Earth Syst. Sci. Data Discuss., https://doi.org/10.5194/essd-2018-7, in review Earth Syst. Sci. Data, in preparation, 2017a2018.

5  Hartten, L., P. Johnston, C. Cox, and D. Wolfe: El Niño Rapid Response (ENRR) Field Campaign: Surface Meteorological Data from Kiritimati Island, January-March 2016 (NCEI Accession 0161526), Version 1.1, NOAA /National Centers for Environmental Information, doi:https://doi.org/10.7289/V51Z42H4, 2017b.

Hosom, D. S., Weller, R. A., Payne, R. E., and Prada, K. E.: The IMET (Improved Meteorology) Ship and Buoy Systems, J.
10  Atmos. Ocean. Tech., 12, 527-540, doi:https://doi.org/10.1175/1520-0426(1995)012<0527:timsab>2.0.co;2, 1995.

Hsu, S. A., Meindl, E. A. and Gilhousen, D. B.: Determining the Power-Law Wind-Profile Exponent under Near-Neutral Stability Conditions at Sea, J. Appl. Meteorol., 33, 757-765, doi:https://doi.org/10.1175/1520-0450(1994)033<0757:DTPLWP>2.0.CO;2, 1994.

Jin, Z., Qiao, Y., Wang, Y., Fang, Y. and Yi, W.: A new parameterization of spectral and broadband ocean albedo, Opt. Express, 19, 26429-26443, https://doi.org/10.1364/OE.19.026429, 2011.

Joint Subcommittee on Ocean Science and Technology (JSOST): Federal Fleet Status Report: Current Capacity and Near-
20  Term Priorities, Washington, D.C., 18, available at http://www.nopp.org/wp-content/uploads/2016/06/federal_fleet_status_report_final_03.2016.pdf, 2016.

Konda, M., Imasato, N., Nishi, K. and Toda, T.: Measurement of the sea surface emissivity, J. Oceanogr., 50, 17-30, https://doi.org/10.1007/BF02233853, 1994.

McClatchey, R. A., Fenn, R. W., Selby, J. E. A., Volz, F. E., and Garing, J. S.: Optical Properties of the Atmosphere (Third Edition), Air Force Cambridge Research Laboratories, Bedford, Mass., Environmental Research Papers, No. 411, 113, available at: www.dtic.mil/dtic/tr/fulltext/u2/753075.pdf, 1972.

30  McFarlane, S. A., Long, C. N., and Flaherty, J. A.: A climatology of surface cloud radiative effects at the ARM Tropical Western Pacific sites, J. Appl. Meteorol. Clim., 52, 996-1012, doi:https://doi.org/10.1175/JAMC-D-12-0189.1, 2013.

Mlawer, E. J., Taubman, S. J., Brown, P. D., Iacono, M. J., and Clough, S. A.: Radiative transfer for inhomogeneous atmospheres: RRTM, a validated correlated-k model for the longwave, J. Geophys. Res.-Atmos., 102, 16663-16682,
35  doi:https://doi.org/10.1029/97JD00237, 1997.

Moat, B. I., Berry, D. I., and Yelland, M. J.: Airflow distortion at instrument sites on the RV *Ronald H. Brown*, Southampton Oceanography Centre, University of Southampton, Southampton, Hampshire, UK, 27, 2001.

40  Morate, O.: 2015 Population and Housing Census Volume 1: Management Report and Basic Tables, National Statistics Office, Ministry of Finance, Bairiki, Tarawa, Republic of Kiribati, 197, available at http://www.mfed.gov.ki/publications/census-report-2015-volume-i-final-report, 2016.

Thoning, K. W., D. R. Kitzis, and A. Crotwell: Atmospheric Carbon Dioxide Dry Air Mole Fractions from quasi-continuous
45  measurements at American Samoa, Version 2016-8, updated annually, NOAA Earth System Research Laboratory (ESRL) Global Monitoring Division (GMD), doi:https://doi.org/10.7289/V51834DB, 2015.

National Centers for Environmental Information (NCEI): Magnetic Field Calculators, https://www.ngdc.noaa.gov/geomag-web/#declination, last access: 18 March 2016.

National Data Buoy Center (NDBC): TAO Mooring Information, http://tao.ndbc.noaa.gov/proj_overview/mooring_ndbc.shtml (last access: 27 March 2017), 2010a.

5 National Data Buoy Center (NDBC): TAO Sensor Specifications, http://tao.ndbc.noaa.gov/proj_overview/sensors_ndbc.shtml (last access: 27 March 2017), 2010b.

Ralph, F. M., Prather, K. A., Cayan, D., Spackman, J. R., DeMott, P., Dettinger, M., Fairall, C., Leung, R., Rosenfeld, D., Rutledge, S., Waliser, D., White, A. B., Cordeira, J., Martin, A., Helly, J., and Intrieri, J.: CalWater field studies designed to quantify the roles of atmospheric rivers and aerosols in modulating U.S. west coast precipitation in a changing climate, B. Am.
10 Meteorol. Soc., 97, 1209-1228, doi:https://doi.org/10.1175/bams-d-14-00043.1, 2016.

Scott, D. A., ed.: A Directory of Wetlands in Oceania, Slimbridge, U.K. and Kuala Lumpur, Malaysia, 370, 1993.

Stamnes, K., Tsay, S. C., Wiscombe, W., and Jayaweera, K.: Numerically stable algorithm for discrete-ordinate-method
15 radiative transfer in multiple scattering and emitting layered media, Appl. Optics, 27, 2502-2509, doi:https://doi.org/10.1364/AO.27.002502, 1988.

20 Vaisala Oyj: DigiCORA® User's Guide, M210488EN-F, Vaisala Oyj, Helsinki, Finland, 113 pp., available from www.vaisala.com, 2010.

Vaisala Oyj: User's Guide: Ground Check Set GC25, Vaisala Oyj, M210329EN-E, Helsinki, Finland, 43 pp., available from www.vaisala.com, 2006.
25
Vaisala Oyj: Vaisala Radiosonde RS92-SGP User's Guide, Vaisala Oyj, M210295EN-J, Helsinki, Finland, 54 pp., available from www.vaisala.com, 2015.

Wang, J., Cole, H. L., Carlson, D. J., Miller, E. R., Beierle, K., Paukkunen, A., and Laine, T. K.: Corrections of Humidity
30 Measurement Errors from the Vaisala RS80 Radiosonde—Application to TOGA COARE Data, Journal of Atmospheric and Oceanic Technology, 19, 981-1002, https://doi.org/10.1175/1520-0426(2002)019<0981:Cohmef>2.0.Co;2, 2002.

Woods Hole Oceanographic Institution (WHOI): ASIMET Module Specifications, http://frodo.whoi.edu/asimet/asimet_module_specs.html#bpr_mod (last access: 31 March 2017), 2010a.
35
Woods Hole Oceanographic Institution (WHOI): ASIMET Sensors, http://www.whoi.edu/instruments/viewInstrument.do?id=14686 (last access: 31 March 2017), 2010b.

World Meteorological Organization: Manual on the Global Observing System, 2010 ed., WMO (Series) no. 544, Secretariat
40 of the World Meteorological Organization, Geneva, Switzerland, 2013.

Zipser, E. J.: The Line Islands Experiment, its place in tropical meteorology and the rise of the fourth school of thought, B. Am. Meteorol. Soc., 51, 1136-1146, 1970.

**Figures**

[Figure]

Figure 1: a) Map of Kiritimati showing the locations of Cassidy International Airport (PLCH) and CXENRR.  Kiritimati map excerpted from U.S. Defense Mapping Agency (DMA) Stock No. 83BHA83130, courtesy of the University of Texas Libraries, The University of Texas at Austin.  Map data from surveys by New Zealand (1938-1941) and the U.S. Navy (to 1962).  Heights are in feet above mean high water springs; soundings are in fathoms (or fathoms and feet if less than 11 fathoms). b) Detail of the CXENRR site at the Captain Cook Hotel.  The bungalow which served as our base of operations is in the center of the image, with markers to its north indicating the initial and final locations of the surface meteorological instruments.

[Figure]

[Figure]

**Figure 2: (a) The surface met station in its final position above the beach at Kiritimati. The anemometer sits atop the assembly, while the temperature/humidity sensor (left) and rain gauge (right) are attached to the crossbeam. Just below the crossbeam is the solar panel, and the box containing the Campbell datalogger, the power supply, and the barometer is mounted just below that. A plastic tube runs from the barometer out the tube on the bottom of the box, and can be seen mounted within a white disk just to the left of the solar panel. (b) The 26 March 2017 00 UTC radiosonde being put on the Campbell box in the shade of the solar panel prior to launch. The pressure tube egress and mount are clearly visible. Photo courtesy of G. Kerber.**

[Figure]

**Figure 3: The original position of the surface instruments (left), and their location after being moved back towards the top of the beach on 5 Feb 2017 (right).**

[Figure]

**Figure 4:** **(a)** SSTs observed by the NOAA Ship *Ronald H. Brown* during ENRR, overlaid on SST anomalies. The latter are the departure from the ERA-Interim (Dee et al., 2011) 1979-2016 February through March mean. **(b) The cruise track of the NOAA Ship *Ronald H. Brown* overlaid on a map showing the location of the TAO buoys visited during its cruise and also the location of Kiritimati Island. The buoys used in the post-processing of the ship's surface pressure, temperature, and humidity are highlighted.**

[revised manuscript text omitted]

---

## Author Comment (AC4) · 25 May 2018

Here is the figure meant to accompany our "Reply to Anonymous Referee #2", which was not included in our reply due to software issues.

Additional information about the data plotted in Figure 1 (originally intended for the caption): Periods during which approximately 25 mm or more rainfall accumulated during a short time are highlighted as "heavy". The ship's latitude and longitude ranges during the legs are as follows: Leg 2, $(2.02° − 5.24°N, 80.50° − 90.65°E)$; Leg 3, $(2.03°S − 4.83°N, 80.49° − 90.79°E)$; Leg 4, $(0.07°S − 3.53°N, 80.5° − 87.91°E)$. Ship data were collected and post-processed into one-minute values by Jim Edson (University of Connecticut), Chris Fairall (NOAA/ESRL/Physical Sciences Division), and Simon deSzoeke (Oregon State University).

**ESSDD**

[Figure]

[Figure]

**Fig. 1.** Ten-minute values of relative humidity at 10 m and leg-accumulated rainfall from three cruise legs during the DYNAMO campaign. See text for details.